# Information-Theoretic Confidence Bounds for Reinforcement Learning

**Xiuyuan Lu**
Stanford University
lxy@stanford.edu

**Benjamin Van Roy**
Stanford University
bvr@stanford.edu

## Abstract

We integrate information-theoretic concepts into the design and analysis of optimistic algorithms and Thompson sampling. By making a connection between information-theoretic quantities and confidence bounds, we obtain results that relate the per-period performance of the agent with its information gain about the environment, thus explicitly characterizing the exploration-exploitation tradeoff. The resulting cumulative regret bound depends on the agent's uncertainty over the environment and quantifies the value of prior information. We show applicability of this approach to several environments, including linear bandits, tabular MDPs, and factored MDPs. These examples demonstrate the potential of a general information-theoretic approach for the design and analysis of reinforcement learning algorithms.

## 1 Introduction

We consider an online decision problem where an agent repeatedly interacts with an uncertain environment and observes outcomes. The agent has a reward function that specifies its preferences over outcomes. The objective of the agent is to sequentially select actions so as to maximize the long-term expected cumulative reward. One classic example is the multi-armed bandit problem, where the agent observes only the reward of the action selected during each period. Another example is episodic reinforcement learning, where the agent selects a policy at the beginning of each episode, and observes a trajectory of states and rewards realized over the episode.

The agent's uncertainty about the environment gives rise to a need to trade off between exploration and exploitation. Exploring parts of the environment that are poorly understood could lead to better performance in the future, while exploiting current knowledge of the environment could lead to higher reward in the short term. Thompson sampling [18] and optimistic algorithms [9, 12] are two classes of algorithms that effectively balance the exploration-exploitation tradeoff and achieve near-optimal performance in many stylized online decision problems. However, most analyses of such algorithms focus on establishing performance guarantees that only exploit the parametric structure of the model [1, 2, 3, 5, 7, 9, 10, 11]. There has not been much focus on how prior information as well as information gain during the learning process affect performance, with few exceptions [15, 16].

In our work, we leverage concepts from information theory to quantify the information gain of the agent and address the exploration-exploitation tradeoff for Thompson sampling and optimistic algorithms. By connecting information-theoretic quantities with confidence bounds, we are able to relate the agent's per-period performance with its information gain about the environment during the period. The relation explicitly characterizes how an algorithm balances exploration and exploitation on a single-period basis. The information gain is represented succinctly using mutual information, which abstracts away from the specific parametric form of the model. The level of abstraction offered by information theory shows promise of these information-theoretic confidence bounds being generalizable to a broad class of problems. Moreover, the resulting cumulative regret bound

explicitly depends on the agent's uncertainty over the environment, which naturally exhibits the value of prior information. We present applications of information-theoretic confidence bounds on three environments, linear bandits, tabular MDPs, and factored MDPs.

One paper that is closely related to our work is [16], which proposes an upper confidence bound (UCB) algorithm for bandit learning with a Gaussian process prior and derives a regret bound that depends on maximal information gain. Some of their results parallel what we establish in the context of the linear bandit, though our analysis extends to Thompson sampling as well. More importantly, our work generalizes the information-theoretic confidence bound approach to problems with significantly more complicated information structure, such as MDPs.

Another closely related paper is [15], which provides an information-theoretic analysis of Thompson sampling for bandit problems with partial feedback. The paper introduces the notion of an information ratio, which relates the one-period regret of Thompson sampling with one-period information gain towards the optimal action. Using this concept, the authors are able to derive a series of regret bounds that depend on the information entropy of the optimal action. While this is an elegant result, it is unclear how to extend the approach to MDPs, as information gain about the optimal policy is hard to quantify. In our paper, we consider information gain about the underlying environment rather than the optimal action or policy, which may be seen as a relaxation of their method. The relaxation allows us to leverage confidence bounds and obtain information-theoretic regret bounds for MDPs.

## 2    Problem formulation

We consider an online decision problem where an agent repeatedly interacts with an uncertain environment and observes outcomes. All random variables are defined on a probability space $(\Omega, \mathcal{F}, \mathbb{P})$. The environment is described by an unknown model parameter $\theta$ which governs the outcome distribution. The agent's uncertainty over the environment is represented as a prior distribution over $\theta$. Thus, $\theta$ will be treated as a random variable in the agent's mind. During each time period $\ell$, the agent selects an action $A_\ell \in \mathcal{A}$ and observes an outcome $Y_{\ell, A_\ell} \in \mathcal{Y}$. We assume that the space of outcomes $\mathcal{Y}$ is a subset of a finite dimensional Euclidean space. Conditioned on the model index $\theta$, outcomes $Y_\ell$ are i.i.d. for $\ell = 1, 2, \ldots$. The agent has a reward function $r : \mathcal{Y} \to \Re$ that encodes its preferences over outcomes. We make a simplifying assumption that rewards are bounded.

**Assumption 1.** *There exists $B \geq 0$ such that* $\sup_{y \in \mathcal{Y}} r(y) - \inf_{y \in \mathcal{Y}} r(y) \leq B$.

The objective of the agent is to maximize its long-term expected cumulative reward. Let $\mathcal{H}_\ell = (A_1, Y_{1, A_1}, \ldots, A_{\ell-1}, Y_{\ell-1, A_{\ell-1}})$ denote the history up to time $\ell$. An algorithm $\pi$ is a sequence of functions $\{\pi_\ell\}_{\ell \geq 1}$ that map histories to distributions over actions. For any $a \in \mathcal{A}$, let $\overline{r}_\theta(a) = \mathbb{E}[r(Y_{1,a})|\theta]$ denote the expected reward of selecting action $a$ under model $\theta$. Let $A^* \in \arg\max_{a \in \mathcal{A}} \overline{r}_\theta(a)$ denote the optimal action under model $\theta$. We define the Bayesian regret of an algorithm $\pi$ over $L$ periods

$$\mathbb{E}[\text{Regret}(L, \pi)] = \sum_{\ell=1}^{L} \mathbb{E}[\overline{r}_\theta(A^*) - \overline{r}_\theta(A_\ell)],$$

where the expectation is taken over the randomness in outcomes, algorithm $\pi$, as well as the prior distribution over $\theta$.

Note that episodic reinforcement learning also falls in the above formulation by considering policies as actions and trajectories as observations.

## 3    Preliminaries

### 3.1    Basic quantities in information theory

For two probability measures $P$ and $Q$ such that $P$ is absolutely continuous with respect to $Q$, the *Kullback-Leibler divergence* between them is

$$D(P||Q) = \int \log \left( \frac{dP}{dQ} \right) dP,$$

where $\frac{dP}{dQ}$ is the Radon-Nikodym derivative of $P$ with respect to $Q$.

Let $P(X) \equiv \mathbb{P}(X \in \cdot)$ denote the probability distribution of a random variable $X$, and let $P(X|Y) \equiv \mathbb{P}(X \in \cdot|Y)$ denote the conditional probability distribution of $X$ conditioned on $Y$.

The *mutual information* between two random variables $X$ and $Y$

$$I(X;Y) = D(P(X,Y) \,\|\, P(X)\,P(Y))$$

is the Kullback-Leibler divergence between their joint distribution and product distribution [6]. $I(X;Y)$ is always nonnegative, and $I(X;Y) = 0$ if and only if $X$ and $Y$ are independent. In our analysis, we will use $I(\theta; A, Y_A)$ to measure the agent's information gain of $\theta$ from selecting an action and observing an outcome.

The conditional mutual information between two random variables $X$ and $Y$, conditioned on a third random variable $Z$, is

$$I(X;Y|Z) = \mathbb{E}[D(P(X,Y|Z) \,\|\, P(X|Z)P(Y|Z))],$$

where the expectation is taken over $Z$. An elegant property of mutual information is that the mutual information between a random variable $X$ and a collection of random variables $Y_1, \dots, Y_n$ can be decomposed into a sum of conditional mutual information using the chain rule.

**Lemma 1.** *(Chain rule of mutual information)*

$$I(X;Y_1, Y_2, \dots, Y_n) = \sum_{i=1}^{n} I(X;Y_i|Y_1, \dots, Y_{i-1}).$$

## 3.2  Notation under posterior distributions

We will use subscript $\ell$ on $\mathbb{P}$ and $\mathbb{E}$ to indicate quantities conditioned on $\mathcal{H}_\ell$, i.e., $\mathbb{P}_\ell(\cdot) \equiv \mathbb{P}(\cdot|\mathcal{H}_\ell) = \mathbb{P}(\cdot|A_1, Y_{1,A_1}, \dots, A_{\ell-1}, Y_{\ell-1,A_{\ell-1}})$, and similarly for $\mathbb{E}_\ell[\cdot]$. Let $P_\ell(X) \equiv \mathbb{P}_\ell(X \in \cdot)$ denote the conditional distribution of a random variable $X$ conditioned on $\mathcal{H}_\ell$. We define filtered mutual information

$$I_\ell(X;Y) = D(P_\ell(X,Y) \,\|\, P_\ell(X)P_\ell(Y)),$$

which is a random variable of $\mathcal{H}_\ell$. Note that by the definition of conditional mutual information,

$$\mathbb{E}[I_\ell(X;Y)] = I(X;Y|\mathcal{H}_\ell) = I(X;Y|A_1, Y_{1,A_1}, \dots, A_{\ell-1}, Y_{\ell-1,A_{\ell-1}}).$$

## 3.3  Algorithms

Thompson sampling is a simple yet effective heuristic for trading off between exploration and exploitation. Conceptually, it samples each action according to the probability that it is optimal. The algorithm typically operates by starting with a prior distribution over $\theta$. During each time period, it samples from the posterior distribution over $\theta$, and selects an action that maximizes the expected reward under the sampled model. It then updates the posterior distribution with the observed outcome.

Another widely studied class of algorithms that effectively trade off between exploration and exploitation are upper confidence bound (UCB) algorithms, which apply the principle of optimism in the face of uncertainty. For each time period, they typically construct an upper confidence bound for the mean reward of each action based on past observations, and then select the action with the highest upper confidence bound.

---

**Algorithm 1** Thompson Sampling

1: **Input**: prior $p$
2: **for** $\ell = 1, 2, \dots, L$ **do**
3:    **Sample**: $\hat{\theta}_\ell \sim p$
4:    **Act**: $A_\ell = \arg\max_{a \in \mathcal{A}} \bar{r}_{\hat{\theta}_\ell}(a)$
5:    **Observe**: $Y_{\ell,A_\ell}$
6:    **Update**: $p \leftarrow \mathbb{P}(\theta \in \cdot|p, A_\ell, Y_{\ell,A_\ell})$
7: **end for**

---

**Algorithm 2** Upper Confidence Bound Algorithm

1: **Input**: upper confidence functions $\{U_\ell\}_{\ell=1}^{L}$
2: **for** $\ell = 1, 2, \dots, L$ **do**
3:    **Act**: $A_\ell = \arg\max_{a \in \mathcal{A}} U_\ell(\mathcal{H}_\ell, a)$
4:    **Observe**: $Y_{\ell,A_\ell}$
5:    **Update**: $\mathcal{H}_{\ell+1} \leftarrow \mathcal{H}_\ell \cup \{A_\ell, Y_{\ell,A_\ell}\}$
6: **end for**

---

# 4 Information-theoretic confidence bounds

Information-theoretic confidence bounds are defined with the intention of capturing the exploration-exploitation tradeoff for Thompson sampling and optimistic algorithms – if the regret is large, the agent must have learned a lot about the environment. Let $\Delta_\ell = \overline{r}_\theta(A^*) - \overline{r}_\theta(A_\ell)$ denote the regret over the $\ell$th period. We aim to obtain per-period regret bound of the form

$$\mathbb{E}_\ell[\Delta_\ell] \leq \Gamma_\ell \sqrt{I_\ell(\theta; A_\ell, Y_{\ell, A_\ell})} + \epsilon_\ell, \tag{1}$$

where $I_\ell(\theta; A_\ell, Y_{\ell, A_\ell})$ is the filtered mutual information between $\theta$ and the action-outcome pair during the $\ell$th period, $\Gamma_\ell$ is the rate at which regret scales with information gain, and we also allow for a small error term $\epsilon_\ell$. If (1) is satisfied with reasonable values for $\Gamma_\ell$ and $\epsilon_\ell$, a large expected regret on the left-hand side would imply that the right-hand side must be large as well, meaning that the agent should gain a lot of information about the environment.

If $\Gamma_\ell$ can be uniformly bounded over $\ell = 1, \ldots, L$, we obtain an information-theoretic regret bound for any algorithm that satisfies (1).

**Proposition 2.** *If* (1) *holds with* $\Gamma_\ell \leq \Gamma$ *for all* $\ell = 1, \ldots, L$, *then*

$$\mathbb{E}[\text{Regret}(L, \pi)] \leq \Gamma \sqrt{L \, I(\theta; A_1, Y_{1, A_1}, \ldots, A_\ell, Y_{\ell, A_\ell})} + \mathbb{E} \sum_{\ell=1}^{L} \epsilon_\ell.$$

The proof follows from Jensen's and the Cauchy-Schwarz inequalities, and the chain rule of mutual information. All complete proofs can be found in the appendix.

The mutual information term on the right-hand side shows how much the agent expects to learn about $\theta$ over $L$ periods. If $\theta$ already concentrates around some value, there is not much to learn, and the result would suggest that the expected regret should be small. In general, the mutual information term can be bounded by the maximal information gain under any algorithm over $L$ periods, though a more careful analysis specialized to the algorithm of interest might lead to a better bound.

One way to obtain a per-period regret bound of the form in Equation (1) is through construction of information-theoretic confidence sets for mean rewards. For each action $a$, the width of the confidence set is designed to depend on the information gain about $\theta$ from observing outcome $Y_{\ell, a}$.

**Lemma 3.** *Under Assumption 1, if*

$$\mathbb{P}_\ell \left( |\overline{r}_\theta(a) - \mathbb{E}_\ell[\overline{r}_\theta(a)]| \leq \frac{\Gamma_\ell}{2} \sqrt{I_\ell(\theta; Y_{\ell, a})} \; \forall a \in \mathcal{A} \right) \geq 1 - \frac{\delta}{2},$$

*then the per-period regret of Thompson sampling and UCB with upper confidence function* $U_\ell(a) = \mathbb{E}_\ell[\overline{r}_\theta(a)] + \frac{\Gamma_\ell}{2} \sqrt{I_\ell(\theta; Y_{\ell, a})}$ *satisfies*

$$\mathbb{E}_\ell[\Delta_\ell] \leq \Gamma_\ell \sqrt{I_\ell(\theta; A_\ell, Y_{\ell, A_\ell})} + \delta B.$$

The proof follows from the probability matching property of Thompson sampling, optimism of UCB, and properties of mutual information.

When the reward function $r(y)$ is Lipschitz continuous, we may alternatively construct confidence sets on outcomes. Further, if the observation noise is additive, we may construct confidence sets on the mean outcomes.

**Lemma 4.** *If* $r(\cdot)$ *is* $K$-*Lipschitz continuous with respect to some norm* $\|\cdot\|$ *on* $\mathcal{Y}$, *and if*

$$\mathbb{P}_\ell \left( \|Y_{\ell, a} - \mathbb{E}_\ell[Y_{\ell, a}]\| \leq \frac{\Gamma_\ell}{2} \sqrt{I_\ell(\theta; Y_{\ell, a})} \; \forall a \in \mathcal{A} \right) \geq 1 - \frac{\delta}{2},$$

*then the per-period regret of Thompson sampling*

$$\mathbb{E}_\ell[\Delta_\ell] \leq K \Gamma_\ell \sqrt{I_\ell(\theta; A_\ell, Y_{\ell, A_\ell})} + \delta B.$$

*Moreover, if there exists a function* $\overline{y} : \Theta \times \mathcal{A} \to \mathcal{Y}$ *such that* $Y_{\ell, a} - \overline{y}(\theta, a)$ *is independent of* $\theta$ *for all* $a \in \mathcal{A}$, *then it is sufficient to have*

$$\mathbb{P}_\ell \left( \|\overline{y}(\theta, a) - \mathbb{E}_\ell[\overline{y}(\theta, a)]\| \leq \frac{\Gamma_\ell}{2} \sqrt{I_\ell(\theta; Y_{\ell, a})} \; \forall a \in \mathcal{A} \right) \geq 1 - \frac{\delta}{2}$$

*for the one-period regret bound to hold.*

An analogous result would hold for a UCB algorithm if outcomes $Y_{\ell,a}$ are scalar valued and the reward function is nondecreasing. We will push further details to the appendix.

## 5 Examples

In this section, we show applications of information-theoretic confidence bounds on linear bandits, tabular MDPs, and factored MDPs. The per-period regret bounds highlight the single-period exploration-exploitation tradeoff for Thompson sampling and the corresponding optimistic algorithms, while the cumulative regret bounds show how the prior distribution over $\theta$ affects regret.

### 5.1 Linear bandits

Let $\mathcal{A} \subset \Re^d$ be a finite action set, and assume that $\sup_{a \in \mathcal{A}} \|a\|_2 \leq 1$. We assume a $N(\mu_1, \Sigma_1)$ prior over the model parameter $\theta \in \Re^d$. At time $\ell$, an action $A_\ell \in \mathcal{A}$ is selected, and $Y_{\ell,A_\ell} = \theta^\top A_\ell + w_\ell$ is observed, where $w_\ell$ are i.i.d. $N(0, \sigma_w^2)$. Conditioned on $\mathcal{H}_\ell$, the posterior distribution of $\theta$ is again normal. Let $\mu_\ell$ and $\Sigma_\ell$ denote the posterior mean and covariance matrix conditioned on $\mathcal{H}_\ell$. We assume that $r(\cdot)$ is bounded under Assumption 1, and is nondecreasing and 1-Lipschitz.

By Lemma 4, since noise is additive, it is sufficient to construct confidence sets on $\overline{Y}_a = \theta^\top a$.

**Lemma 5.** *Under the assumptions stated in Section 5.1,*

$$\mathbb{P}_\ell \left( \left| \overline{Y}_a - \mathbb{E}_\ell \overline{Y}_a \right| \leq \frac{\Gamma_\ell}{2} \sqrt{I_\ell(\theta; Y_{\ell,a})} \ \forall a \in \mathcal{A} \right) \geq 1 - \frac{\delta}{2},$$

*for*

$$\Gamma_\ell = 4 \sqrt{\frac{\sigma_{\ell,\max}^2}{\log\left(1 + \frac{\sigma_{\ell,\max}^2}{\sigma_w^2}\right)} \log \frac{4|\mathcal{A}|}{\delta}}, \quad \text{where} \ \sigma_{\ell,\max}^2 = \max_{a \in \mathcal{A}} a^\top \Sigma_\ell a.$$

Thus, it follows from Lemma 4 that the one-period regret of Thompson sampling and UCB with $U_\ell(a) = \mathbb{E}_\ell \overline{Y}_a + \frac{\Gamma_\ell}{2} \sqrt{I_\ell(\theta; Y_{\ell,a})}$ satisfies

$$\mathbb{E}_\ell[\Delta_\ell] \leq \Gamma_\ell \sqrt{I_\ell(\theta; A_\ell, Y_{\ell,A_\ell})} + \delta B.$$

By Proposition 2, we have the following Bayesian regret bound for Thompson sampling and UCB by choosing $\delta = \frac{1}{L}$.

**Proposition 6.** *Under the assumptions stated in Section 5.1, the Bayesian regret of Thompson sampling and UCB over $L$ periods is*

$$\mathbb{E}[\text{Regret}(L, \pi)] \leq \Gamma \sqrt{L \ I(\theta; A_1, Y_{1,A_1}, \dots, A_L, Y_{L,A_L})} + B$$

*where*

$$\Gamma = 4 \sqrt{\frac{\sigma_{1,\max}^2}{\log\left(1 + \frac{\sigma_{1,\max}^2}{\sigma_w^2}\right)} \log(4|\mathcal{A}|L)}.$$

The following lemma bounds the maximal information gain over $L$ time periods.

**Lemma 7.** *For any $\mathcal{H}_\ell$-adapted action sequence,*

$$I(\theta; A_1, Y_{1,A_1}, \dots, A_L, Y_{L,A_L}) \leq \frac{1}{2} d \log\left(1 + \frac{\lambda_{\max} L}{\sigma_w^2}\right),$$

*where $\lambda_{\max}$ is the largest eigenvalue of $\Sigma_1$.*

It follows from Lemma 7 that the Bayesian regret of Thompson sampling and UCB is bounded by $O(\sqrt{dL \log |\mathcal{A}|} \log L)$, which matches the result in [16]. For large action sets, it is possible to apply a discretization argument and obtain a regret bound of order $O(d\sqrt{L} \log L)$.

## 5.2 Tabular MDPs

We consider the problem of learning to optimize a random finite-horizon MDP $M = (\mathcal{S}, \mathcal{A}, R, P, \tau, \rho)$ in repeated episodes. $\mathcal{S}$ is the state space, $\mathcal{A}$ is the action space, and we assume that both $\mathcal{S}$ and $\mathcal{A}$ are finite. Assume that for each $s, a$, the reward distribution is Bernoulli with mean $R(s, a)$, where $R(s, a)$ follows an independent Beta prior with parameter $\alpha_{1,s,a}^R \in \Re^2$. We further assume that for each $s, a$, the transition distribution $P(s, a, \cdot)$ follows an independent Dirichlet prior with parameter $\alpha_{1,s,a}^P \in \Re^{|\mathcal{S}|}$. $\tau$ is a fixed time horizon, and $\rho$ is the initial state distribution. We make the following simplifying assumption on the prior parameters.

**Assumption 2.** *For all $s \in \mathcal{S}$ and $a \in \mathcal{A}$, $\alpha_{1,s,a}^R(i) \geq 1$ for $i \in \{1, 2\}$, and $\alpha_{1,s,a}^P(j) \geq \frac{2}{|\mathcal{S}|}$ for all $j \in \{1, \ldots, |\mathcal{S}|\}$.*

A (deterministic) policy $\mu$ is a sequence of functions $(\mu^0, \ldots, \mu^{\tau-1})$ that map states to actions. During each episode $\ell$, the agent selects a policy $\mu_\ell$, and observes a trajectory

$$Y_{\ell,\mu_\ell} = (s_{\ell,0}, a_{\ell,0}, r_{\ell,1}, s_{\ell,1}, \ldots, s_{\ell,\tau-1}, a_{\ell,\tau-1}, r_{\ell,\tau}).$$

We define the value function of a policy $\mu$ under an MDP $\tilde{M}$

$$V_{\mu,k}^{\tilde{M}}(s) := \mathbb{E}\left[ \sum_{t=k}^{\tau-1} R(s_t, a_t) \,\middle|\, M = \tilde{M}, \mu, s_k = s \right].$$

Define the expected value of a policy $\mu$ under an MDP $\tilde{M}$

$$\overline{V}_\mu^{\tilde{M}} = \mathbb{E}\left[ V_{\mu,0}^{\tilde{M}}(s_0) \,\middle|\, M = \tilde{M}, \mu \right].$$

Let $\mu^*$ denote an optimal policy for the true environment $M$. The Bayesian regret of an algorithm $\pi$ over $L$ episodes is

$$\mathbb{E}[\text{Regret}(L, \pi)] = \sum_{\ell=1}^{L} \mathbb{E}\left[ \overline{V}_{\mu^*}^M - \overline{V}_{\mu_\ell}^M \right],$$

where the expectation is taken over the randomness in observations, algorithm $\pi$, as well as the prior distribution over $M$.

We will construct confidence bounds on value functions in the spirit of Lemma 3. As we will see in the following two lemmas, the structure of MDPs allows us to break down the deviation of value functions and the information gain at the level of state-action pairs. Thus, it would be sufficient to construct confidence sets for the reward and transition functions for individual state-action pairs.

The following lemma decomposes the planning error to a sum of on-policy Bellman errors. The proof can be found in Section 5.1 of [13].

**Lemma 8.** *For any MDP $\hat{M}$ and policy $\mu$,*

$$\overline{V}_\mu^{\hat{M}} - \overline{V}_\mu^M = \sum_{t=0}^{\tau-1} \mathbb{E}\left[ \left( \hat{R}(s_t, a_t) - R(s_t, a_t) \right) + \left( \hat{P}(s_t, a_t) - P(s_t, a_t) \right)^\top V_{\mu,t+1}^{\hat{M}} \,\middle|\, \hat{M}, M, \mu \right],$$

*where $\hat{R}$, $\hat{P}$ are the reward and transition functions under $\hat{M}$.*

Let $\mathcal{H}_{\ell t} = (\mu_\ell, s_{\ell,0}, a_{\ell,0}, \ldots, r_{\ell,t}, s_{\ell,t})$ denote the history of episode $\ell$ up to time $t$ and the policy selected for episode $\ell$. The chain rule of mutual information gives us the following lemma.

**Lemma 9.** *(information decomposition)*

$$I_\ell\left(M; \mu_\ell, Y_{\ell,\mu_\ell}\right) = \sum_{t=0}^{\tau-1} I_\ell\left(M; (s_{\ell,t}, a_{\ell,t}, r_{\ell,t+1}, s_{\ell,t+1}) \,\middle|\, \mathcal{H}_{\ell t}\right).$$

By Lemmas 8 and 9, we will construct information-theoretic confidence sets for reward and transition distributions individually for each state-action pair.

**Lemma 10.** *Let $r_{\ell,t,s,a} \sim \text{Bernoulli}(R(s,a)) \,|\, \mathcal{H}_\ell, \mathcal{H}_{\ell t}$ and $s'_{\ell,t,s,a} \sim P(s,a,\cdot) \,|\, \mathcal{H}_\ell, \mathcal{H}_{\ell t}$. Then,*

$$\mathbb{P}_\ell \left( |R(s,a) - \mathbb{E}_\ell R(s,a)| \leq \Gamma^R \sqrt{\min_{\tilde{t},h_{\tilde{t}}} I_\ell(R; s, a, r_{\ell,\tilde{t},s,a} \,|\, \mathcal{H}_{\ell\tilde{t}} = h_{\tilde{t}})} \right) \geq 1 - \delta \qquad (2)$$

*and*

$$\mathbb{P}_\ell \left( \left| \left(P(s,a) - \mathbb{E}_\ell P(s,a)\right)^\top V^M_{\mu^*,t+1} \right| \leq \Gamma^P \sqrt{\min_{\tilde{t},h_{\tilde{t}}} I_\ell(P; s, a, s'_{\ell,\tilde{t},s,a} \,|\, \mathcal{H}_{\ell\tilde{t}} = h_{\tilde{t}})} \right) \geq 1 - \delta \quad (3)$$

*for all $t$ and all $s, a$ such that $\mathbf{1}^\top \alpha^R_{\ell,s,a} \geq \tau - 1$ and $\mathbf{1}^\top \alpha^P_{\ell,s,a} \geq \tau - 1$, respectively, where*

$$\Gamma^R = \sqrt{24 \log \tfrac{2}{\delta}} \quad \text{and} \quad \Gamma^P = \tau \sqrt{24 \log \tfrac{2}{\delta}}.$$

The terms $\min_{\tilde{t},h_{\tilde{t}}} I_\ell(R; s, a, r_{\ell,\tilde{t},s,a} \,|\, \mathcal{H}_{\ell\tilde{t}} = h_{\tilde{t}})$ and $\min_{\tilde{t},h_{\tilde{t}}} I_\ell(P; s, a, s'_{\ell,\tilde{t},s,a} \,|\, \mathcal{H}_{\ell\tilde{t}} = h_{\tilde{t}})$ measure the minimum per-step information gain that the agent can obtain about the reward and transition functions of a state-action pair during the $\ell$th episode, conditioned on $\mathcal{H}_\ell$, where the minimum is taken over all possible values of the time step $0 \leq \tilde{t} \leq \tau - 1$ and all possible realizations of the trajectory $\mathcal{H}_{\ell\tilde{t}}$.

Lemma 10 allows us to construct a high probability confidence set $\mathcal{M}_\ell$ over $M$, which is discussed more in details in the appendix. The corresponding UCB algorithm will select $\mu_\ell = \arg\max_\mu \max_{\hat{M} \in \mathcal{M}_\ell} \overline{V}^{\hat{M}}_\mu$. Combining with Lemmas 8 and 9, we are able to obtain a per-period regret bound for Thompson sampling and UCB in the form of (1), with $\Gamma_\ell = \tilde{O}(\tau\sqrt{\tau})$. This leads to the following proposition.

**Proposition 11.** *The Bayesian regret of Thompson sampling and UCB over $L$ episodes is*

$$\mathbb{E}[\text{Regret}(L, \pi)] = \tilde{O}\left( \tau \sqrt{\tau L \, I(M; \mu_1, Y_{1,\mu_1}, \dots, \mu_L, Y_{L,\mu_L})} \right).$$

We show in the appendix that $I(M; \mu_1, Y_{1,\mu_1}, \dots, \mu_L, Y_{L,\mu_L}) = \tilde{O}(S^2 A)$. Though we conjecture that a bound of $\tilde{O}(SA)$ may be attainable under appropriate conditions. The conjecture is supported by our simulations, as discussed in the appendix.

## 5.3 Factored MDPs

Factored MDPs [4] are a class of structured MDPs where transitions are represented by a dynamic Bayesian network [8], and can typically be encoded in a compact parametric form. Our information-theoretic approach will lead to a regret bound for Thompson sampling and UCB that depends on the prior of the model parameter, which typically has dimension exponentially smaller than the state space and action space.

We start with some definitions common in the literature [17].

**Definition 1.** *Let $\mathcal{X} = \mathcal{X}_1 \times \cdots \times \mathcal{X}_d$ be a factored set. For any subset of indices $Z \subseteq \{1, 2, \dots, d\}$, we define the scope set $\mathcal{X}[Z] := \otimes_{i \in Z} \mathcal{X}_i$. Further, for any $x \in \mathcal{X}$, define the scope variable $x[Z] \in \mathcal{X}[Z]$ to be the value of the variables $x_i \in \mathcal{X}_i$ with indices $i \in Z$. If $Z$ is a singleton, we will write $x[i]$ for $x[\{i\}]$.*

Let $\mathcal{P}(\mathcal{X}, \mathcal{Y})$ denote the set of functions that map $x \in \mathcal{X}$ to a probability distribution on $\mathcal{Y}$.

**Definition 2.** *The reward function class $\mathcal{R} \subset \mathcal{P}(\mathcal{X}, \Re)$ is factored over $\mathcal{S} \times \mathcal{A} = \mathcal{X} = \mathcal{X}_1 \times \cdots \times \mathcal{X}_d$ with scopes $Z_1, \dots, Z_m$ if and only if, for all $R \in \mathcal{R}$, $x \in \mathcal{X}$, there exist functions $\{R_i \in \mathcal{P}(\mathcal{X}[Z_i], \Re)\}_{i=1}^m$ such that*

$$\mathbb{E}[r] = \sum_{i=1}^m \mathbb{E}[r_i]$$

*where $r \sim R(x)$ is equal to $\sum_{i=1}^m r_i$ with each $r_i \sim R_i(x[Z_i])$ individually observed.*

**Definition 3.** *The transition function class $\mathcal{P} \subset \mathcal{P}(\mathcal{X}, \mathcal{S})$ is factored over $\mathcal{S} \times \mathcal{A} = \mathcal{X} = \mathcal{X}_1 \times \cdots \times \mathcal{X}_d$ and $\mathcal{S} = \mathcal{S}_1 \times \cdots \times \mathcal{S}_n$ with scopes $Z_1, \ldots, Z_n$ if and only if, for all $P \in \mathcal{P}$, $x \in \mathcal{X}$, $s \in \mathcal{S}$, there exist functions $\{P_j \in \mathcal{P}(\mathcal{X}[Z_j], \mathcal{S}_j)\}_{j=1}^{n}$ such that*

$$P(s|x) = \prod_{j=1}^{n} P_j(s[j] \mid x[Z_j]).$$

A factored MDP is an MDP with factored rewards and transitions. Let $\mathcal{X} = \mathcal{S} \times \mathcal{A}$. A factored MDP is fully characterized by

$$M = \left( \{\mathcal{X}_i\}_{i=1}^{d}; \ \{Z_i^R\}_{i=1}^{m}; \ \{R_i\}_{i=1}^{m}; \ \{\mathcal{S}_j\}_{j=1}^{n}; \ \{Z_j^P\}_{j=1}^{n}; \ \{P_j\}_{j=1}^{n}; \ \tau; \ \rho \right),$$

where $\{Z_i^R\}_{i=1}^{m}$ and $\{Z_j^P\}_{j=1}^{n}$ are the scopes for the reward and transition functions, which we assume are known to the agent, $\tau$ is a fixed time horizon, and $\rho$ is the initial state distribution.

We assume that $|Z_i^R|, |Z_j^P| \leq \zeta \ll d$ and $|\mathcal{X}_i| \leq K$, so the domain of any reward and transition function has size at most $K^\zeta$. Let $D_R = \sum_{i=1}^{m} \left|\mathcal{X}[Z_i^R]\right|$ and $D_P = \sum_{j=1}^{n} \left|\mathcal{X}[Z_j^P]\right|$ be the sum of the cardinality of the domains, and let $D = D_R + D_P \ll |\mathcal{S}||\mathcal{A}|$. To simplify exposition, let us assume that $|\mathcal{S}_j| \leq K$ for all $j = 1, \ldots, n$.

Let $x_i^R$ denote an element in $\mathcal{X}[Z_i^R]$, and $x_j^P$ denote an element in $\mathcal{X}[Z_j^P]$. We assume that the factorized rewards are Bernoulli. With a slight abuse of notation, we let $R_i(x_i^R)$ denote the mean reward, and we assume that $R_i(x_i^R) \sim \text{Beta}(\alpha_{1,i,x_i^R}^R)$, where $\alpha_{1,i,x_i^R}^R \in \Re^2$. We further assume that $P_j(x_j^P, \cdot) \sim \text{Dirichlet}(\alpha_{1,j,x_j^P}^P)$, $\alpha_{1,j,x_j^P}^P \in \Re^{|\mathcal{S}_j|}$. Similar to the tabular case, we assume that each component of $\alpha_{1,i,x_i^R}^R$ is at least 1, and each component of $\alpha_{1,j,x_j^P}^P$ is at least $\frac{2}{|\mathcal{S}_j|}$ for all $i$ and $j$.

Lemmas 8 and 9 still hold. Since the reward and transition functions can be factorized, we will construct information-theoretic confidence bounds on these factor functions.

**Lemma 12.** *Let $r_{\ell,t,i,x_i^R} \sim \text{Bernoulli}(R(x_i^R)) \mid \mathcal{H}_\ell, \mathcal{H}_{\ell t}$ and $s'_{\ell,t,j,x_j^P} \sim P_j(x_j^P, \cdot) \mid \mathcal{H}_\ell, \mathcal{H}_{\ell t}$. Then,*

$$\mathbb{P}_\ell \left( \left| R_i(x_i^R) - \mathbb{E}_\ell R_i(x_i^R) \right| \leq \Gamma^R \sqrt{\min_{\tilde{t}, h_{\tilde{t}}} I_\ell(R_i; x_i^R, r_{\ell, \tilde{t}, i, x_i^R} \mid \mathcal{H}_{\ell \tilde{t}} = h_{\tilde{t}})} \right) \geq 1 - \delta \qquad (4)$$

*and*

$$\mathbb{P}_\ell \left( \|P_j(x_j^P) - \mathbb{E}_\ell P_j(x_j^P)\|_1 \leq \Gamma_j^P \sqrt{\min_{\tilde{t}, h_{\tilde{t}}} I_\ell(P_j; x_j^P, s'_{\ell, \tilde{t}, j, x_j^P} \mid \mathcal{H}_{\ell \tilde{t}} = h_{\tilde{t}})} \right) \geq 1 - \delta \qquad (5)$$

*for all $s, a$ such that $\mathbf{1}^\top \alpha_{\ell,i,x_i^R}^R \geq \tau - 1$ and $\mathbf{1}^\top \alpha_{\ell,j,x_j^P}^P \geq \tau - 1$, respectively, with*

$$\Gamma^R = 2\sqrt{6 \log \tfrac{2}{\delta}} \quad and \quad \Gamma_j^P = 4\sqrt{6|\mathcal{S}_j| \log \tfrac{2}{\delta}}.$$

The lemma allows us to obtain a per-period regret bound in the form of (1) for Thompson sampling and a UCB algorithm that uses (4) and (5) to construct confidence sets. As shown in the appendix, the scaling factor $\Gamma_\ell = \tilde{O}(m\tau\sqrt{(m+n)K\tau})$, which leads to the following proposition.

**Proposition 13.** *The Bayesian regret of Thompson sampling and UCB over $L$ episodes is*

$$\mathbb{E}[\text{Regret}(L, \pi)] = \tilde{O}\left( m\tau \sqrt{(m+n)K\tau L \, I\left(M; \mu_1, Y_{1,\mu_1}, \ldots, \mu_L, Y_{L,\mu_L}\right)} \right).$$

Similar to the tabular case, we show that $I\left(M; \mu_1, Y_{1,\mu_1}, \ldots, \mu_L, Y_{L,\mu_L}\right)$ is $\tilde{O}(KD)$ for any algorithm, while we conjecture that this may be $\tilde{O}(D)$ under appropriate conditions. The resulting regret bound matches the one in [14] if the conjecture proves to be true. Again, our bound in Proposition 13 reveals an explicit dependence on the prior uncertainty, which is not captured by previous work.

# 6   Conclusion

We introduce information-theoretic confidence bounds for analyzing Thompson sampling and deriving optimistic algorithms. We show that the information-theoretic approach allows us to formally quantify the agent's information gain of the unknown environment, and to explicitly characterize the exploration-exploitation tradeoff for linear bandits, tabular MDPs, and factored MDPs. This work opens up multiple directions for future research. It would be interesting to extend information-theoretic confidence bounds to a broader range of problems and see whether a general information-theoretic framework is plausible for addressing online decision problems. It would also be interesting to think about whether an information-theoretic perspective could lead to tighter regret bounds for Thompson sampling and optimistic algorithms. One may also consider the practical implications of these confidence bounds and how they can be used to design better reinforcement learning algorithms.

**Acknowledgments**

This work was generously supported by the Charles and Katherine Lin Graduate Fellowship, the Dantzig-Lieberman Operations Research Fellowship, and a research grant from Boeing. We would also like to thank Ayfer Özgür and Daniel Russo for helpful discussions.

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
