[Supplementary Material]

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

*Proof.* We have

$$
\begin{aligned}
\mathbb{E}[\text{Regret}(L, \pi)] &= \mathbb{E} \sum_{\ell=1}^{L} \mathbb{E}_\ell[\Delta_\ell] \\
&\leq \sum_{\ell=1}^{L} \mathbb{E}\Gamma_\ell \sqrt{I_\ell(\theta; A_\ell, Y_{\ell,A_\ell})} + \epsilon_\ell \\
&\leq \Gamma \left( \sum_{\ell=1}^{L} \mathbb{E}\sqrt{I_\ell(\theta; A_\ell, Y_{\ell,A_\ell})} \right) + \mathbb{E} \sum_{\ell=1}^{L} \epsilon_\ell \\
&\leq \Gamma \left( \sum_{\ell=1}^{L} \sqrt{\mathbb{E}I_\ell(\theta; A_\ell, Y_{\ell,A_\ell})} \right) + \mathbb{E} \sum_{\ell=1}^{L} \epsilon_\ell \\
&\leq \Gamma \sqrt{L \sum_{\ell=1}^{L} \mathbb{E}I_\ell(\theta; A_\ell, Y_{\ell,A_\ell})} + \mathbb{E} \sum_{\ell=1}^{L} \epsilon_\ell \\
&= \Gamma \sqrt{L \sum_{\ell=1}^{L} I(\theta; A_\ell, Y_{\ell,A_\ell} | A_1, Y_{1,A_1}, \ldots, A_{\ell-1}, Y_{\ell-1,A_{\ell-1}})} + \mathbb{E} \sum_{\ell=1}^{L} \epsilon_\ell \\
&= \Gamma \sqrt{LI(\theta; A_1, Y_{1,A_1}, \ldots, A_L, Y_{L,A_L})} + \mathbb{E} \sum_{\ell=1}^{L} \epsilon_\ell,
\end{aligned}
$$

where the first two inequalities follow from our assumptions, the third inequality follows from Jensen's inequality, the fourth inequality follows from Cauchy-Schwarz, and the last equality follows from the chain rule of mutual information. $\square$

**Lemma 3.** *Under Assumption 1, if*

$$\mathbb{P}_\ell \left( |\bar{r}_\theta(a) - \mathbb{E}_\ell\left[\bar{r}_\theta(a)\right]| \leq \frac{\Gamma_\ell}{2} \sqrt{I_\ell(\theta; Y_{\ell,a})} \ \forall a \in \mathcal{A} \right) \geq 1 - \frac{\delta}{2},$$

*then the per-period regret of Thompson sampling and UCB with upper confidence function* $U_\ell(a) = \mathbb{E}_\ell\left[\bar{r}_\theta(a)\right] + \frac{\Gamma_\ell}{2}\sqrt{I_\ell(\theta; Y_{\ell,a})}$ *satisfies*

$$\mathbb{E}_\ell\left[\Delta_\ell\right] \leq \Gamma_\ell \sqrt{I_\ell(\theta; A_\ell, Y_{\ell,A_\ell})} + \delta B.$$

*Proof.* By the probability matching property of Thompson sampling,

$$\mathbb{E}_\ell[\Delta_\ell] = \mathbb{E}_\ell\left[\bar{r}_\theta(A^*) - \bar{r}_\theta(A_\ell)\right] = \mathbb{E}_\ell\left[\bar{r}_{\hat{\theta}_\ell}(A_\ell) - \bar{r}_\theta(A_\ell)\right],$$

where $\hat{\theta}_\ell$ is the parameter sampled by Thompson sampling for period $\ell$. Define

$$\Theta_\ell = \left\{ \bar{\theta} \in \Theta : \left|\bar{r}_{\bar{\theta}}(a) - \mathbb{E}_\ell\left[\bar{r}_\theta(a)\right]\right| \leq \frac{\Gamma_\ell}{2}\sqrt{I_\ell(\theta; Y_{\ell,a})} \ \forall a \in \mathcal{A} \right\}.$$

The probability matching property then implies that $\mathbb{P}_\ell(\hat{\theta}_\ell \in \Theta_\ell) = \mathbb{P}_\ell(\theta \in \Theta_\ell) \geq 1 - \delta/2$. Thus,

$$
\begin{aligned}
\mathbb{E}_\ell[\Delta_\ell] &\leq \mathbb{E}_\ell[\mathbf{1}_{\theta,\hat{\theta}_\ell \in \Theta_\ell}(\bar{r}_{\hat{\theta}_\ell}(A_\ell) - \bar{r}_\theta(A_\ell))] + \delta B \\
&\leq \mathbb{E}_\ell\left[\Gamma_\ell \sum_{a \in \mathcal{A}} \mathbf{1}_{A_\ell = a} \sqrt{I_\ell(\theta; Y_{\ell,a})}\right] + \delta B \\
&\leq \Gamma_\ell \sum_{a \in \mathcal{A}} \mathbb{P}_\ell(A_\ell = a)\sqrt{I_\ell(\theta; Y_{\ell,a})} + \delta B \\
&\leq \Gamma_\ell \sqrt{\sum_{a \in \mathcal{A}} \mathbb{P}_\ell(A_\ell = a) I_\ell(\theta; Y_{\ell,a})} + \delta B \\
&= \Gamma_\ell \sqrt{\sum_{a \in \mathcal{A}} \mathbb{P}_\ell(A_\ell = a) I_\ell(\theta; Y_{\ell,A_\ell}|A_\ell = a)} + \delta B \\
&= \Gamma_\ell \sqrt{I_\ell(\theta; Y_{\ell,A_\ell}|A_\ell)} + \delta B \\
&= \Gamma_\ell \sqrt{I_\ell(\theta; A_\ell, Y_{\ell,A_\ell})} + \delta B,
\end{aligned}
$$

where the first and the last equalities follow from the conditional independence between $A_\ell$ and $\theta$ conditioned on $\mathcal{H}_\ell$. Therefore, the one-period regret of Thompson sampling is

$$
\mathbb{E}_\ell[\Delta_\ell] \leq \Gamma_\ell \sqrt{I_\ell(\theta; A_\ell, Y_{\ell,A_\ell})} + \delta B.
$$

For a UCB algorithm with upper confidence functions $U_\ell(a) = \mathbb{E}_\ell[\bar{r}_\theta(a)] + \frac{\Gamma_\ell}{2}\sqrt{I_\ell(\theta; Y_{\ell,a})}$, we have

$$
\begin{aligned}
\mathbb{E}_\ell[\Delta_\ell] &\leq \mathbb{E}_\ell[\mathbf{1}_{\theta \in \Theta_\ell}(\bar{r}_\theta(A^*) - \bar{r}_\theta(A_\ell))] + \frac{1}{2}\delta B \\
&\leq \mathbb{E}_\ell[\mathbf{1}_{\theta \in \Theta_\ell}(\bar{r}_\theta(A^*) - U_\ell(A^*) + U_\ell(A_\ell) - \bar{r}_\theta(A_\ell))] + \frac{1}{2}\delta B \\
&\leq \mathbb{E}_\ell[\mathbf{1}_{\theta \in \Theta_\ell}(U_\ell(A_\ell) - \bar{r}_\theta(A_\ell))] + \frac{1}{2}\delta B \\
&\leq \mathbb{E}_\ell\left[\Gamma_\ell \sum_{a \in \mathcal{A}} \mathbf{1}_{A_\ell = a}\sqrt{I_\ell(\theta; Y_{\ell,a})}\right] + \frac{1}{2}\delta B.
\end{aligned}
$$

The rest of the proof follows similarly. $\qquad\qquad\square$

**Lemma 4.** *If $r(\cdot)$ is $K$-Lipschitz continuous with respect to some norm $\|\cdot\|$ on $\mathcal{Y}$, and if*

$$
\mathbb{P}_\ell\left(\|Y_{\ell,a} - \mathbb{E}_\ell[Y_{\ell,a}]\| \leq \frac{\Gamma_\ell}{2}\sqrt{I_\ell(\theta; Y_{\ell,a})} \ \forall a \in \mathcal{A}\right) \geq 1 - \frac{\delta}{2},
$$

*then the per-period regret of Thompson sampling*

$$
\mathbb{E}_\ell[\Delta_\ell] \leq K\Gamma_\ell\sqrt{I_\ell(\theta; A_\ell, Y_{\ell,A_\ell})} + \delta B.
$$

*Moreover, if there exists a function $\bar{y} : \Theta \times \mathcal{A} \to \mathcal{Y}$ such that $Y_{\ell,a} - \bar{y}(\theta, a)$ is independent of $\theta$ for all $a \in \mathcal{A}$, then it is sufficient to have*

$$
\mathbb{P}_\ell\left(\|\bar{y}(\theta, a) - \mathbb{E}_\ell[\bar{y}(\theta, a)]\| \leq \frac{\Gamma_\ell}{2}\sqrt{I_\ell(\theta; Y_{\ell,a})} \ \forall a \in \mathcal{A}\right) \geq 1 - \frac{\delta}{2}
$$

*for the one-period regret bound to hold.*

*Proof.* Let $\hat{\theta}_\ell$ be the parameter sampled by Thompson sampling during period $\ell$. Let $\hat{Y}_\ell$ be a random variable drawn from the outcome distribution indexed by $\hat{\theta}_\ell$. Define $\mathcal{Y}_\ell = \{y_\ell \in \mathcal{Y}^{|\mathcal{A}|} : \|y_{\ell,a} - $

$\mathbb{E}_\ell\left[Y_{\ell,a}\right]\| \leq \frac{\Gamma_\ell}{2}\sqrt{I_\ell(\theta;Y_{\ell,a})}\ \forall a \in \mathcal{A}\}$. Since $\hat{Y}_\ell$ and $Y_\ell$ are identically distributed conditioned on $\mathcal{H}_\ell$, we have $\mathbb{P}_\ell(Y_\ell \in \mathcal{Y}_\ell) = \mathbb{P}_\ell(\hat{Y}_\ell \in \mathcal{Y}_\ell) \geq 1 - \delta/2$. Thus,

$$
\begin{aligned}
\mathbb{E}_\ell[\Delta_\ell] &= \mathbb{E}_\ell\left[\bar{r}_{\hat{\theta}_\ell}(A_\ell) - \bar{r}_\theta(A_\ell)\right]\\
&= \mathbb{E}_\ell[r(\hat{Y}_{\ell,A_\ell}) - r(Y_{\ell,A_\ell})]\\
&\leq \mathbb{E}_\ell\left[\mathbf{1}_{Y_\ell,\hat{Y}_\ell\in\mathcal{Y}_\ell}\left(r(\hat{Y}_{\ell,A_\ell}) - r(Y_{\ell,A_\ell})\right)\right] + \delta B\\
&\leq K\mathbb{E}_\ell\left[\mathbf{1}_{Y_\ell,\hat{Y}_\ell\in\mathcal{Y}_\ell}\|\hat{Y}_{\ell,A_\ell} - Y_{\ell,A_\ell}\|\right] + \delta B.
\end{aligned}
$$

On $Y_\ell,\hat{Y}_\ell \in \mathcal{Y}_\ell$, $\|\hat{Y}_{\ell,A_\ell} - Y_{\ell,A_\ell}\| \leq \Gamma_\ell\sum_{a\in\mathcal{A}}\mathbf{1}_{A_\ell=a}\sqrt{I_\ell(\theta;Y_{\ell,a})}$. The rest of the analysis follows similarly to the proof of Lemma 3.

Now we show that if the observation noise is additive, it is sufficient to construct confidence sets around mean outcomes. Suppose that there exists a function $\bar{y} : \Theta \times \mathcal{A} \to \mathcal{Y}$ such that $w_{\ell,a} = Y_{\ell,a} - \bar{y}(\theta,a)$ is independent of $\theta$. Let $\hat{Y}_{\ell,a} = \bar{y}(\hat{\theta}_\ell,a) + w_{\ell,a}$. Then $(\theta,Y_\ell)$ and $(\hat{\theta}_\ell,\hat{Y}_\ell)$ are identically distributed conditioned on $\mathcal{H}_\ell$. Let $\Theta_\ell = \{\tilde{\theta} \in \Theta : \|\bar{y}(\tilde{\theta},a) - \mathbb{E}_\ell\left[\bar{y}(\theta,a)\right]\| \leq \frac{\Gamma_\ell}{2}\sqrt{I_\ell(\theta;Y_{\ell,a})}\ \forall a \in \mathcal{A}\}$. We have

$$
\begin{aligned}
\mathbb{E}_\ell[\Delta_\ell] &= \mathbb{E}_\ell\left[\bar{r}_{\hat{\theta}_\ell}(A_\ell) - \bar{r}_\theta(A_\ell)\right]\\
&= \mathbb{E}_\ell[r(\hat{Y}_{\ell,A_\ell}) - r(Y_{\ell,A_\ell})]\\
&\leq \mathbb{E}_\ell\left[\mathbf{1}_{\theta,\hat{\theta}_\ell\in\Theta_\ell}\left(r(\hat{Y}_{\ell,A_\ell}) - r(Y_{\ell,A_\ell})\right)\right] + \delta B\\
&\leq K\mathbb{E}_\ell\left[\mathbf{1}_{\theta,\hat{\theta}_\ell\in\Theta_\ell}\|\hat{Y}_{\ell,A_\ell} - Y_{\ell,A_\ell}\|\right] + \delta B\\
&= K\mathbb{E}_\ell\left[\mathbf{1}_{\theta,\hat{\theta}_\ell\in\Theta_\ell}\|\bar{y}(\hat{\theta}_\ell,A_\ell) - \bar{y}(\theta,A_\ell)\|\right] + \delta B.
\end{aligned}
$$

The rest of the proof follows similarly. $\square$

If outcomes are scalar-valued and $r(\cdot)$ is nondecreasing, then a similar result holds for a UCB algorithm with upper confidence functions

$$
U_\ell(a) = \mathbb{E}_\ell[Y_{\ell,a}] + \frac{\Gamma_\ell}{2}\sqrt{I_\ell(\theta;Y_{\ell,a})},
$$

and moreover if the observation noise is additive and independent of $\theta$ and actions,

$$
U_\ell(a) = \mathbb{E}_\ell[\bar{y}(\theta,a)] + \frac{\Gamma_\ell}{2}\sqrt{I_\ell(\theta;Y_{\ell,a})}.
$$

To see this, note that

$$
\begin{aligned}
\mathbb{E}_\ell[\Delta_\ell] &= \mathbb{E}_\ell\left[r(Y_{\ell,A^*}) - r(Y_{\ell,A_\ell})\right]\\
&\leq \mathbb{E}_\ell\left[\mathbf{1}_{Y_\ell\in\mathcal{Y}_\ell}(r(Y_{\ell,A^*}) - r(Y_{\ell,A_\ell}))\right] + \frac{1}{2}\delta B\\
&\leq \mathbb{E}_\ell\left[\mathbf{1}_{Y_\ell\in\mathcal{Y}_\ell}(r(Y_{\ell,A^*}) - r(U_\ell(A^*)) + r(U_\ell(A_\ell)) - r(Y_{\ell,A_\ell}))\right] + \frac{1}{2}\delta B\\
&\leq \mathbb{E}_\ell\left[\mathbf{1}_{Y_\ell\in\mathcal{Y}_\ell}(r(U_\ell(A_\ell)) - r(Y_{\ell,A_\ell}))\right] + \frac{1}{2}\delta B\\
&\leq K\mathbb{E}_\ell\left[\mathbf{1}_{Y_\ell\in\mathcal{Y}_\ell}(U_\ell(A_\ell) - Y_{\ell,A_\ell})\right] + \frac{1}{2}\delta B\\
&\leq K\mathbb{E}_\ell\left[\Gamma_\ell\sum_{a\in\mathcal{A}}\mathbf{1}_{A_\ell=a}\sqrt{I_\ell(\theta;Y_{\ell,a})}\right] + \frac{1}{2}\delta B.
\end{aligned}
$$

The rest of the proof is similar to the proof of Lemma 3.

If the observation noise $w_\ell$ is additive and independent of $\theta$ and actions, we have

$$
\begin{aligned}
\mathbb{E}_\ell[\Delta_\ell] &= \mathbb{E}_\ell\left[r(Y_{\ell,A^*}) - r(Y_{\ell,A_\ell})\right] \\
&= \mathbb{E}_\ell\left[r(\overline{y}(\theta, A^*) + w_\ell) - r(\overline{y}(\theta, A_\ell) + w_\ell)\right] \\
&\leq \mathbb{E}_\ell\left[\mathbf{1}_{\theta\in\Theta_\ell}(r(\overline{y}(\theta, A^*) + w_\ell) - r(\overline{y}(\theta, A_\ell) + w_\ell))\right] + \frac{1}{2}\delta B \\
&\leq \mathbb{E}_\ell\Big[\mathbf{1}_{\theta\in\Theta_\ell}\big(r(\overline{y}(\theta, A^*) + w_\ell) - r(U_\ell(A^*) + w_\ell) \\
&\qquad\qquad + r(U_\ell(A_\ell) + w_\ell) - r(\overline{y}(\theta, A_\ell) + w_\ell))\Big] + \frac{1}{2}\delta B \\
&\leq \mathbb{E}_\ell\left[\mathbf{1}_{\theta\in\Theta_\ell}(r(U_\ell(A_\ell) + w_\ell) - r(\overline{y}(\theta, A_\ell) + w_\ell))\right] + \frac{1}{2}\delta B \\
&\leq K\mathbb{E}_\ell\left[\mathbf{1}_{\theta\in\Theta_\ell}(U_\ell(A_\ell) - \overline{y}(\theta, A_\ell))\right] + \frac{1}{2}\delta B \\
&\leq K\mathbb{E}_\ell\left[\Gamma_\ell \sum_{a\in\mathcal{A}} \mathbf{1}_{A_\ell=a}\sqrt{I_\ell(\theta; Y_{\ell,a})}\right] + \frac{1}{2}\delta B \\
&\leq K\Gamma_\ell\sqrt{I_\ell(\theta; A_\ell, Y_{\ell,A_\ell})} + \frac{1}{2}\delta B.
\end{aligned}
$$

## B  Linear bandits

The following lemma gives a formula for the mutual information between a normal random variable and a linear observation corrupted by gaussian noise. We use $h(\cdot)$ to denote the differential entropy of a continuous random variable.

**Lemma 14.** *If $\theta \in \Re^d$ follows $N(\mu, \Sigma)$, and $Y = a^\top\theta + w$ where $a \in \Re^d$ is fixed and $w \sim N(0, \sigma_w^2)$, then*

$$
I(\theta; Y) = \frac{1}{2}\log\left(1 + \frac{a^\top\Sigma a}{\sigma_w^2}\right).
$$

*Proof.* We have

$$
\begin{aligned}
I(\theta; Y) &= h(\theta) - h(\theta|Y) \\
&= \frac{1}{2}\log\det(2\pi e\Sigma) - \frac{1}{2}\log\det\left(2\pi e\left(\Sigma^{-1} + \frac{aa^\top}{\sigma_w^2}\right)^{-1}\right) \\
&= \frac{1}{2}\log\det\left(I_d + \frac{\Sigma aa^\top}{\sigma_w^2}\right) \\
&= \frac{1}{2}\log\left(1 + \frac{a^\top\Sigma a}{\sigma_w^2}\right),
\end{aligned}
$$

where the last step follows from Sylvester's determinant theorem. □

Since noise is additive in a linear bandit, by Lemma 4, it is sufficient to construct confidence sets around mean outcomes $\overline{Y}_a = a^\top\theta$.

**Lemma 5.** *Under the assumptions stated in Section 5.1,*

$$
\mathbb{P}_\ell\left(\left|\overline{Y}_a - \mathbb{E}_\ell\overline{Y}_a\right| \leq \frac{\Gamma_\ell}{2}\sqrt{I_\ell(\theta; Y_{\ell,a})} \ \forall a \in \mathcal{A}\right) \geq 1 - \frac{\delta}{2},
$$

*for*

$$
\Gamma_\ell = 4\sqrt{\frac{\sigma_{\ell,\max}^2}{\log\left(1 + \frac{\sigma_{\ell,\max}^2}{\sigma_w^2}\right)}\log\frac{4|\mathcal{A}|}{\delta}}, \quad \text{where } \sigma_{\ell,\max}^2 = \max_{a\in\mathcal{A}} a^\top\Sigma_\ell a.
$$

*Proof.* Note that $\overline{Y}_a = a^\top \theta$ is distributed as $N(a^\top \mu_\ell, a^\top \Sigma_\ell a)$ conditioned on $\mathcal{H}_\ell$. By the Chernoff bound,

$$
\begin{aligned}
\mathbb{P}_\ell \left( |\overline{Y}_a - \mathbb{E}_\ell \overline{Y}_a| \geq \frac{\Gamma_\ell}{2} \sqrt{I_\ell(\theta; Y_{\ell,a})} \right) &\leq 2 \exp\left( -\frac{\left( \frac{\Gamma_\ell}{2} \sqrt{I_\ell(\theta; Y_{\ell,a})} \right)^2}{2 a^\top \Sigma_\ell a} \right) \\
&= 2 \exp\left( -\frac{\Gamma_\ell^2 I_\ell(\theta; Y_{\ell,a})}{8 a^\top \Sigma_\ell a} \right) \\
&\leq 2 \exp\left( -\frac{\sigma_{\ell,\max}^2}{\log\left(1 + \frac{\sigma_{\ell,\max}^2}{\sigma_w^2}\right)} \frac{\log\left(1 + \frac{a^\top \Sigma_\ell a}{\sigma_w^2}\right)}{a^\top \Sigma_\ell a} \log\frac{4|\mathcal{A}|}{\delta} \right) \\
&\leq \frac{\delta}{2|\mathcal{A}|},
\end{aligned}
$$

where the last inequality follows from the monotonicity of $\frac{x}{\log(1+x)}$ for $x > 0$ and the fact that $a^\top \Sigma_\ell a \leq \sigma_{\ell,\max}^2$. Applying a union bound over actions gives

$$
\mathbb{P}_\ell \left( |\overline{Y}_a - \mathbb{E}_\ell \overline{Y}_a| \leq \frac{\Gamma_\ell}{2} \sqrt{I_\ell(\theta; Y_{\ell,a})} \ \forall a \in \mathcal{A} \right) \geq 1 - \frac{\delta}{2}.
$$

$\square$

**Proposition 6.** *Under the assumptions stated in Section 5.1, the Bayesian regret of Thompson sampling and UCB over L periods is*

$$
\mathbb{E}[\mathrm{Regret}(L, \pi)] \leq \Gamma \sqrt{L\, I(\theta; A_1, Y_{1,A_1}, \ldots, A_L, Y_{L,A_L})} + B
$$

*where*

$$
\Gamma = 4 \sqrt{\frac{\sigma_{1,\max}^2}{\log\left(1 + \frac{\sigma_{1,\max}^2}{\sigma_w^2}\right)} \log(4|\mathcal{A}|L)}.
$$

*Proof.* The result follows from Proposition 2, Lemma 4, and Lemma 5 by taking $\delta = \frac{1}{L}$. $\square$

**Lemma 7.** *For any $\mathcal{H}_\ell$-adapted action sequence,*

$$
I(\theta; A_1, Y_{1,A_1}, \ldots, A_L, Y_{L,A_L}) \leq \frac{1}{2} d \log\left(1 + \frac{\lambda_{\max} L}{\sigma_w^2}\right),
$$

*where $\lambda_{\max}$ is the largest eigenvalue of $\Sigma_1$.*

*Proof.* Let $\{A_\ell\}_{\ell=1}^L$ be any $\mathcal{H}_\ell$-adaptive action sequence. Let $h(\cdot)$ denote the differential entropy. We have

$$
\begin{aligned}
I(\theta; A_1, Y_{1,A_1}, \ldots, A_L, Y_{L,A_L}) &= h(\theta) - h(\theta | A_1, Y_{1,A_1}, \ldots, A_L, Y_{L,A_L}) \\
&= \frac{1}{2} \log \det(2\pi e \Sigma_1) - \frac{1}{2} \mathbb{E}[\log \det(2\pi e \Sigma_{L+1})] \\
&= \frac{1}{2} \mathbb{E}\left[ \log \frac{\det(\Sigma_1)}{\det(\Sigma_{L+1})} \right].
\end{aligned}
$$

Let $\lambda_k(A)$ denote the $k^{\text{th}}$ largest eigenvalue of a matrix $A$. Let $V = \frac{1}{\sigma_w^2} \sum_{\ell=1}^L A_\ell A_\ell^\top$. Since $\Sigma_{L+1}^{-1} = \Sigma_1^{-1} + V$ where both $\Sigma_1^{-1}$ and $V$ are symmetric, we have

$$
\lambda_k(\Sigma_{L+1}^{-1}) \leq \lambda_k(\Sigma_1^{-1}) + \lambda_1(V) \quad \text{for } k = 1, \ldots, d.
$$

Further, since $V$ is positive semi-definite,

$$
\lambda_1(V) \leq \sum_{k=1}^d \lambda_k(V) = \mathrm{tr}(V) \leq \frac{L}{\sigma_w^2}.
$$

Thus,

$$\log\left(\det(\Sigma_1)\det(\Sigma_{L+1}^{-1})\right) \le \log\left(\prod_{k=1}^{d}\lambda_i(\Sigma_1)\left(\frac{1}{\lambda_i(\Sigma_1)}+\frac{L}{\sigma_w^2}\right)\right) \le d\log\left(1+\frac{\lambda_{\max}L}{\sigma_w^2}\right).$$

Therefore,

$$I(\theta;A_1,Y_{1,A_1},\ldots,A_L,Y_{L,A_L}) \le \frac{1}{2}d\log\left(1+\frac{\lambda_{\max}L}{\sigma_w^2}\right).$$

$\square$

## C   Tabular MDPs

The following lemma gives a formula for the mutual information between a Dirichlet random variable and a Categorical observation.

**Lemma 15.** *If $p \sim Dirichlet(\alpha)$ for some $\alpha \in \Re_+^N$, and $Y$ is drawn from distribution $p$, then*

$$I(p;Y) = \sum_{i=1}^{N}\frac{\alpha_i}{\overline{\alpha}}\left(\psi(\alpha_i+1)-\log\alpha_i\right)-\left(\psi(\overline{\alpha}+1)-\log\overline{\alpha}\right),$$

*where $\overline{\alpha} = \mathbf{1}^\top\alpha$ and $\psi(\cdot)$ is the digamma function. Further, if $\alpha_i \ge 2/N$ for all $i = 1,\ldots,N$, then*

$$I(p;Y) \ge \frac{1}{6\overline{\alpha}}.$$

*Proof.* The differential entropy of a Dirichlet random variable is

$$h(p) = \log B(\alpha) + (\overline{\alpha}-N)\psi(\overline{\alpha}) - \sum_{j=1}^{N}(\alpha_j-1)\psi(\alpha_j),$$

where $B(\cdot)$ is the multivariate Beta function and $\psi(\cdot)$ is the digamma function. Then,

$$
\begin{aligned}
I(p;Y) &= h(p)-h(p|Y)\\
&= \left(\log B(\alpha)+(\overline{\alpha}-N)\psi(\overline{\alpha})-\sum_{j=1}^{N}(\alpha_j-1)\psi(\alpha_j)\right)\\
&\quad -\sum_{i=1}^{N}\frac{\alpha_i}{\overline{\alpha}}\left(\log B(\alpha+e_i)+(\overline{\alpha}+1-N)\psi(\overline{\alpha}+1)-\sum_{j\neq i}(\alpha_j-1)\psi(\alpha_j)-\alpha_i\psi(\alpha_i+1)\right)\\
&= \sum_{i=1}^{N}\frac{\alpha_i}{\overline{\alpha}}\left(\psi(\alpha_i+1)-\log\alpha_i\right)-\left(\psi(\overline{\alpha}+1)-\log\overline{\alpha}\right)
\end{aligned}
$$

after simplifications. Moreover, if $\alpha_i \ge 2/N$ for all $i = 1,\ldots,N$, we have $I(p;Y) \ge \frac{1}{6\overline{\alpha}}$ using the digamma inequalities stated below in Lemma 16.  $\square$

**Lemma 16.** *(Digamma inequalities) For $x > 0$,*

$$\log(x+\tfrac{1}{2}) \le \psi(x+1) \le \log x + \frac{1}{2x}.$$

Our confidence sets rely on the stochastic dominance results established in [15] and [16]. For completeness, we restate the definition and results below.

**Definition 4.** *(Stochastic optimism) Let $X$ and $Y$ be real-valued random variables with finite expectation. We say that $X$ is stochastically optimistic for $Y$ if for any convex and increasing $u : \Re \to \Re$,*

$$\mathbb{E}[u(X)] \ge \mathbb{E}[u(Y)].$$

*We will write $X \succeq_{\mathrm{so}} Y$ for this relation.*

**Lemma 17.** *(Gaussian-Dirichlet dominance) For all fixed $v \in [0,1]^N$, $\alpha \in [0,\infty)^N$ with $\overline{\alpha} = \mathbf{1}^\top \alpha \geq 2$, if $X = p^\top v$ for $p \sim Dirichlet(\alpha)$ and $Y \sim N(\alpha^\top v/\overline{\alpha}, 1/\overline{\alpha})$, then $\mathbb{E}[X] = \mathbb{E}[Y]$ and $Y \succeq_{so} X$.*

One implication of Lemma 17 is that $p^\top v$ will have sub-Gaussian tails with sub-Gaussian parameter $1/\overline{\alpha}$. Now we are ready to construct confidence sets.

**Lemma 10.** *Let $r_{\ell,t,s,a} \sim \mathrm{Bernoulli}(R(s,a)) \mid \mathcal{H}_\ell, \mathcal{H}_{\ell t}$ and $s'_{\ell,t,s,a} \sim P(s,a,\cdot) \mid \mathcal{H}_\ell, \mathcal{H}_{\ell t}$. Then,*

$$\mathbb{P}_\ell \left( |R(s,a) - \mathbb{E}_\ell R(s,a)| \leq \Gamma^R \sqrt{\min_{\tilde{t}, h_{\tilde{t}}} I_\ell(R; s, a, r_{\ell,\tilde{t},s,a} \mid \mathcal{H}_{\ell\tilde{t}} = h_{\tilde{t}})} \right) \geq 1 - \delta \qquad (2)$$

*and*

$$\mathbb{P}_\ell \left( \left| (P(s,a) - \mathbb{E}_\ell P(s,a))^\top V^M_{\mu^*,t+1} \right| \leq \Gamma^P \sqrt{\min_{\tilde{t}, h_{\tilde{t}}} I_\ell(P; s, a, s'_{\ell,\tilde{t},s,a} \mid \mathcal{H}_{\ell\tilde{t}} = h_{\tilde{t}})} \right) \geq 1 - \delta \quad (3)$$

*for all $t$ and all $s, a$ such that $\mathbf{1}^\top \alpha^R_{\ell,s,a} \geq \tau - 1$ and $\mathbf{1}^\top \alpha^P_{\ell,s,a} \geq \tau - 1$, respectively, where*

$$\Gamma^R = \sqrt{24 \log \tfrac{2}{\delta}} \quad and \quad \Gamma^P = \tau \sqrt{24 \log \tfrac{2}{\delta}}.$$

*Proof.* Let $\alpha^R_{\ell,t,s,a}$ and $\alpha^P_{\ell,t,s,a}$ denote the posterior parameters for the reward and transition functions associated with state-action pair $(s,a)$ conditioned on $\mathcal{H}_\ell$ and $\mathcal{H}_{\ell t}$. Let $\overline{\alpha}^R_{\ell,s,a} = \mathbf{1}^\top \alpha^R_{\ell,s,a}$, $\overline{\alpha}^P_{\ell,s,a} = \mathbf{1}^\top \alpha^P_{\ell,s,a}$, and similarly, $\overline{\alpha}^R_{\ell,t,s,a} = \mathbf{1}^\top \alpha^R_{\ell,t,s,a}$, $\overline{\alpha}^P_{\ell,t,s,a} = \mathbf{1}^\top \alpha^P_{\ell,t,s,a}$.

If $\overline{\alpha}^R_{\ell,s,a} \geq \tau - 1$, then for any $0 \leq \tilde{t} \leq \tau - 1$, $\overline{\alpha}^R_{\ell,\tilde{t},s,a} \leq \overline{\alpha}^R_{\ell,s,a} + \tau - 1 \leq 2\overline{\alpha}^R_{\ell,s,a}$. Then, by Lemma 15, we have

$$\frac{1}{\overline{\alpha}^R_{\ell,s,a}} \leq \frac{2}{\overline{\alpha}^R_{\ell,\tilde{t},s,a}} \leq 12 I_\ell \left( R; s, a, r_{\ell,\tilde{t},s,a} \mid \mathcal{H}_{\ell\tilde{t}} = h_{\tilde{t}} \right)$$

for any $0 \leq \tilde{t} \leq \tau - 1$ and trajectory $h_{\tilde{t}}$. Similarly, $\overline{\alpha}^P_{\ell,s,a} \geq \tau - 1$ implies that

$$\frac{1}{\overline{\alpha}^P_{\ell,s,a}} \leq \frac{2}{\overline{\alpha}^P_{\ell,\tilde{t},s,a}} \leq 12 I_\ell \left( P; s, a, s'_{\ell,\tilde{t},s,a} \mid \mathcal{H}_{\ell\tilde{t}} = h_{\tilde{t}} \right)$$

for any $0 \leq \tilde{t} \leq \tau - 1$ and $h_{\tilde{t}}$.

Then, by Lemma 17, we have for $\overline{\alpha}^R_{\ell,s,a} \geq \tau - 1$,

$$
\begin{aligned}
1 - \delta &\leq \mathbb{P}_\ell \left( |R(s,a) - \mathbb{E}_\ell R(s,a)| \leq \sqrt{\frac{2}{\overline{\alpha}^R_{\ell,s,a}} \log \frac{2}{\delta}} \right) \\
&\leq \mathbb{P}_\ell \left( |R(s,a) - \mathbb{E}_\ell R(s,a)| \leq \Gamma^R \sqrt{\min_{\tilde{t}, h_{\tilde{t}}} I_\ell \left( R; s, a, r_{\ell,\tilde{t},s,a} \mid \mathcal{H}_{\ell\tilde{t}} = h_{\tilde{t}} \right)} \right).
\end{aligned}
$$

To obtain concentration around transitions, we cannot directly apply Lemma 17, since $V^M_{\mu^*,t+1}$ is correlated with $P(s,a)$. We will use results in [16] that guarantee sub-Gaussian behavior despite the correlation. By Lemma 3 in [16], we have for $\overline{\alpha}^P_{\ell,s,a} \geq \tau - 1$ and any $t$,

$$
\begin{aligned}
1 - \delta &\leq \mathbb{P}_\ell \left( \left| (P(s,a) - \mathbb{E}_\ell P(s,a))^\top V^M_{\mu^*,t+1} \right| \leq \tau \sqrt{\frac{2}{\overline{\alpha}^P_{\ell,s,a}} \log \frac{2}{\delta}} \right) \\
&\leq \mathbb{P}_\ell \left( \left| (P(s,a) - \mathbb{E}_\ell P(s,a))^\top V^M_{\mu^*,t+1} \right| \leq \Gamma^P \sqrt{\min_{\tilde{t}, h_{\tilde{t}}} I_\ell \left( P; s, a, s'_{\ell,\tilde{t},s,a} \mid \mathcal{H}_{\ell\tilde{t}} = h_{\tilde{t}} \right)} \right).
\end{aligned}
$$

$\square$

We now construct confidence sets around MDPs. Let $\overline{R}_\ell$ and $\overline{P}_\ell$ denote the posterior mean of reward and transition functions conditioned on $\mathcal{H}_\ell$, and define shorthands

$$I_\ell^{\min}(R(s,a)) \equiv \min_{\tilde{t},h_{\tilde{t}}} I_\ell\left(R; s,a,r_{\ell,\tilde{t},s,a} \mid \mathcal{H}_{\ell\tilde{t}} = h_{\tilde{t}}\right),$$

$$I_\ell^{\min}(P(s,a)) \equiv \min_{\tilde{t},h_{\tilde{t}}} I_\ell\left(P; s,a,s'_{\ell,\tilde{t},s,a} \mid \mathcal{H}_{\ell\tilde{t}} = h_{\tilde{t}}\right).$$

Define confidence set

$$\mathcal{M}_\ell = \left\{ \tilde{M} : \left|\tilde{R}(s,a) - \overline{R}_\ell(s,a)\right| \leq \Gamma^R \sqrt{I_\ell^{\min}(R(s,a))} \ \forall s,a \text{ s.t. } \overline{\alpha}^R_{\ell,s,a} \geq \tau - 1, \text{ and} \right.$$

$$\left. \left|\left(\tilde{P}(s,a) - \overline{P}_\ell(s,a)\right)^\top V^{\tilde{M}}_{\tilde{\mu},t+1}\right| \leq \Gamma^P \sqrt{I_\ell^{\min}(P(s,a))} \ \forall t,s,a \text{ s.t. } \overline{\alpha}^P_{\ell,s,a} \geq \tau - 1 \right\},$$
(6)

where $\tilde{\mu}$ is a greedy policy with respect to $\tilde{M}$, and

$$\Gamma^R = \sqrt{24\log\frac{8|\mathcal{S}||\mathcal{A}|\tau}{\delta}} \quad \text{and} \quad \Gamma^P = \tau\sqrt{24\log\frac{8|\mathcal{S}||\mathcal{A}|\tau}{\delta}}.$$
(7)

By Lemma 10 and the union bound, we have $\mathbb{P}_\ell(M \in \mathcal{M}_\ell) \geq 1 - \delta/2$.

Together with Lemmas 8 and 9, we have the following result.

**Proposition 11.** *The Bayesian regret of Thompson sampling and UCB over L episodes is*

$$\mathbb{E}[\text{Regret}(L,\pi)] = \tilde{O}\left(\tau\sqrt{\tau L \ I(M;\mu_1,Y_{1,\mu_1},\ldots,\mu_L,Y_{L,\mu_L})}\right).$$

*Proof.* We will show that the one-period regret of Thompson sampling and UCB is

$$\mathbb{E}_\ell[\Delta_\ell] \leq \Gamma\sqrt{I_\ell(M;\mu_\ell,Y_{\ell,\mu_\ell})} + \epsilon_\ell$$
(8)

with

$$\Gamma = 4(\tau+1)\sqrt{6\tau\log\frac{8|\mathcal{S}||\mathcal{A}|\tau}{\delta}}$$

and

$$\epsilon_\ell = \tau\mathbb{E}_\ell\left[\sum_{t=0}^{\tau-1}\mathbf{1}(n_{\ell,s_{\ell,t},a_{\ell,t}} < \tau - 3)\right] + \delta\tau,$$

where $n_{\ell,s,a}$ is the number of times $(s,a)$ has been visited up to episode $\ell$. The proposition then follows from Proposition 2 by letting $\delta = \frac{1}{L}$ and noting that

$$\mathbb{E}\sum_{\ell=1}^L \epsilon_\ell = \mathbb{E}\sum_{\ell=1}^L\left[\tau\left(\sum_{t=0}^{\tau-1}\mathbf{1}(n_{\ell,s_{\ell,t},a_{\ell,t}} < \tau - 3)\right) + \delta\tau\right] \leq |\mathcal{S}||\mathcal{A}|\tau^2 + \delta\tau L.$$

We first show that (8) holds for Thompson sampling. Let $\hat{M}_\ell$ denote the MDP sampled by Thompson sampling during episode $\ell$. By the probability matching property, $\mathbb{P}_\ell(\hat{M}_\ell \in \mathcal{M}_\ell) = \mathbb{P}_\ell(M \in \mathcal{M}_\ell) \geq 1 - \frac{\delta}{2}$, where $\mathcal{M}_\ell$ is defined in (6), Moreover, an argument similar to Lemma 10 gives

$$\mathbb{P}_\ell\left( \left|R(s,a) - \overline{R}_\ell(s,a)\right| \leq \Gamma^R\sqrt{I_\ell^{\min}(R(s,a))} \ \forall s,a \text{ s.t. } \overline{\alpha}^R_{\ell,s,a} \geq \tau - 1, \text{ and} \right.$$

$$\left. \left|(P(s,a) - \overline{P}_\ell(s,a))^\top V^{\hat{M}_\ell}_{\mu_\ell,t+1}\right| \leq \Gamma^P\sqrt{I_\ell^{\min}(P(s,a))} \ \forall t,s,a \text{ s.t. } \overline{\alpha}^P_{\ell,s,a} \geq \tau - 1 \right) \geq 1 - \frac{\delta}{2},$$

with $\Gamma^R$ and $\Gamma^P$ are defined in (7). Let $E_\ell$ denote the intersection of the above event and $\{\hat{M}_\ell \in \mathcal{M}_\ell\}$. Then $\mathbb{P}_\ell(E_\ell) \geq 1 - \delta$.

Denote $x_{\ell,t} \equiv (s_{\ell,t}, a_{\ell,t})$. By the probability matching property and Lemma 8, we have

$$
\begin{aligned}
\mathbb{E}_\ell\left[\Delta_\ell\right] &= \mathbb{E}_\ell\left[\overline{V}_{\mu^*}^M - \overline{V}_{\mu_\ell}^M\right] \\
&= \mathbb{E}_\ell\left[\overline{V}_{\mu_\ell}^{\hat{M}_\ell} - \overline{V}_{\mu_\ell}^M\right] \\
&\leq \mathbb{E}_\ell\left[\mathbf{1}_{E_\ell}\left(\overline{V}_{\mu_\ell}^{\hat{M}_\ell} - \overline{V}_{\mu_\ell}^M\right)\right] + \delta\tau \\
&= \mathbb{E}_\ell\left[\mathbf{1}_{E_\ell} \sum_{t=0}^{\tau-1} \mathbb{E}_\ell\left[\left(\hat{R}_\ell(x_{\ell,t}) - R(x_{\ell,t})\right) + \left(\hat{P}_\ell(x_{\ell,t}) - P(x_{\ell,t})\right)^\top V_{\mu_\ell,t+1}^{\hat{M}_\ell}\,\Big|\,\hat{M}_\ell, M, \mu_\ell\right]\right] + \delta\tau \\
&= \sum_{t=0}^{\tau-1} \mathbb{E}_\ell\left[\mathbf{1}_{E_\ell}\left(\left(\hat{R}_\ell(x_{\ell,t}) - R(x_{\ell,t})\right) + \left(\hat{P}_\ell(x_{\ell,t}) - P(x_{\ell,t})\right)^\top V_{\mu_\ell,t+1}^{\hat{M}_\ell}\right)\right] + \delta\tau \\
&\leq \sum_{t=0}^{\tau-1} \mathbb{E}_\ell\left[\sum_{s,a} \mathbf{1}(x_{\ell,t} = (s,a))\left(2\Gamma^R \sqrt{I_\ell^{\min}(R(s,a))} + 2\Gamma^P \sqrt{I_\ell^{\min}(P(s,a))}\right)\right] \\
&\qquad + \sum_{t=0}^{\tau-1} \mathbb{E}_\ell\left[\mathbf{1}\left(\overline{\alpha}_{\ell,x_{\ell,t}}^R < \tau-1\right) + (\tau-1)\mathbf{1}\left(\overline{\alpha}_{\ell,x_{\ell,t}}^P < \tau-1\right)\right] + \delta\tau \\
&\leq 2\sum_{t=0}^{\tau-1} \mathbb{E}_\ell\left[\sum_{s,a} \mathbf{1}(x_{\ell,t} = (s,a))\left(\Gamma^R \sqrt{I_\ell^{\min}(R(s,a))} + \Gamma^P \sqrt{I_\ell^{\min}(P(s,a))}\right)\right] + \epsilon_\ell.
\end{aligned}
$$

Let $\mathbb{P}_{\ell t}(\cdot) = \mathbb{P}(\cdot|\mathcal{H}_\ell, \mathcal{H}_{\ell t})$, and let $I_{\ell t}(X;Y)$ denote the mutual information under the base measure $\mathbb{P}_{\ell t}(\cdot)$. By definition,

$$
I_\ell^{\min}(R(s,a)) \leq I_{\ell t}(R; s, a, r_{\ell,t,s,a}) \quad \text{and} \quad I_\ell^{\min}(P(s,a)) \leq I_{\ell t}(P; s, a, s'_{\ell,t,s,a}).
$$

Moreover,

$$
\Gamma^R\sqrt{I_{\ell t}(R; s, a, r_{\ell,t,s,a})} + \Gamma^P\sqrt{I_{\ell t}(P; s, a, s'_{\ell,t,s,a})} \leq (\Gamma^R + \Gamma^P)\sqrt{I_{\ell t}(M; s, a, r_{\ell,t,s,a}, s'_{\ell,t,s,a})},
$$

and thus,

$$
\Gamma^R\sqrt{I_\ell^{\min}(R(s,a))} + \Gamma^P\sqrt{I_\ell^{\min}(P(s,a))} \leq (\Gamma_\ell^R + \Gamma_\ell^P)\sqrt{I_{\ell t}(M; s, a, r_{\ell,t,s,a}, s'_{\ell,t,s,a})}.
$$

Therefore, we have

$$
\begin{aligned}
\mathbb{E}_\ell\left[\Delta_\ell\right] &\leq 2(\Gamma^R + \Gamma^P)\sum_{t=0}^{\tau-1} \mathbb{E}_\ell\left[\sum_{s,a} \mathbf{1}(x_{\ell,t} = (s,a))\sqrt{I_{\ell t}(M; s, a, r_{\ell,t,s,a}, s'_{\ell,t,s,a})}\right] + \epsilon_\ell \\
&= 2(\Gamma^R + \Gamma^P)\sum_{t=0}^{\tau-1} \mathbb{E}_\ell\left[\sqrt{I_{\ell t}(M; s_{\ell,t}, a_{\ell,t}, r_{\ell,t+1}, s_{\ell,t+1})}\right] + \epsilon_\ell \\
&\leq 2(\Gamma^R + \Gamma^P)\sqrt{\tau \sum_{t=0}^{\tau-1} \mathbb{E}_\ell I_{\ell t}(M; s_{\ell,t}, a_{\ell,t}, r_{\ell,t+1}, s_{\ell,t+1})} + \epsilon_\ell \\
&= 2(\Gamma^R + \Gamma^P)\sqrt{\tau \sum_{t=0}^{\tau-1} I_\ell(M;\ s_{\ell,t}, a_{\ell,t}, r_{\ell,t+1}, s_{\ell,t+1}\mid\mathcal{H}_{\ell t})} + \epsilon_\ell \\
&= 2(\Gamma^R + \Gamma^P)\sqrt{\tau I_\ell(M;\ \mu_\ell, Y_{\ell,\mu_\ell})} + \epsilon_\ell,
\end{aligned}
$$

where the second inequality follows from Jensen's inequality and Cauchy-Schwarz inequality, and the last step follows from the information decomposition stated in Lemma 9. Define

$$
\Gamma = 2\sqrt{\tau}(\Gamma^R + \Gamma^P) = 4(\tau+1)\sqrt{6\tau \log\frac{8|\mathcal{S}||\mathcal{A}|\tau}{\delta}},
$$

Figure 1: Scaling of mutual information between a Dirichlet random variable and categorical observations with respect to (a) number of categories $N$, and (b) number of observations (fixing $N = 800$).

and we obtain (8).

Now we show that UCB with confidence sets $\mathcal{M}_\ell$ defined by (6) also satisfies (8). Let $\mu_\ell = \arg\max_\mu \max_{\hat{M} \in \mathcal{M}_\ell} \overline{V}_\mu^{\hat{M}}$, and let $\hat{M}_\ell$ denote the optimistic MDP that corresponds to $\mu_\ell$. Again, by Lemmas 17 and 10 and the union bound,

$$\mathbb{P}_\ell \left( \left| \left( P(s,a) - \overline{P}_\ell(s,a) \right)^\top V_{\mu_\ell, t+1}^{\hat{M}_\ell} \right| \leq \Gamma^P \sqrt{I_\ell^{\min}(P(s,a))} \; \forall t, s, a \text{ s.t. } \overline{\alpha}_{\ell,s,a}^P \geq \tau - 1 \right) \geq 1 - \frac{\delta}{2},$$

Let $E_\ell$ denote the intersection of the above event and $\{M \in \mathcal{M}_\ell\}$. Then $\mathbb{P}_\ell(E_\ell) \geq 1 - \delta$. We have

$$\mathbb{E}_\ell \left[ \Delta_\ell \right] = \mathbb{E}_\ell \left[ \overline{V}_{\mu^*}^M - \overline{V}_{\mu_\ell}^M \right] \leq \mathbb{E}_\ell \left[ \mathbf{1}_{E_\ell} \left( \overline{V}_{\mu^*}^M - \overline{V}_{\mu_\ell}^M \right) \right] + \delta\tau \leq \mathbb{E}_\ell \left[ \mathbf{1}_{E_\ell} \left( \overline{V}_{\mu_\ell}^{\hat{M}_\ell} - \overline{V}_{\mu_\ell}^M \right) \right] + \delta\tau.$$

The rest of the analysis follows similarly. □

Let $T = \tau L$ be the total number of time steps. We provide a bound on the maximal information gain of order $O\left( |\mathcal{S}|^2 |\mathcal{A}| \log \frac{T}{|\mathcal{S}||\mathcal{A}|} \right)$. Though we conjecture that a bound of order $\tilde{O}(|\mathcal{S}||\mathcal{A}|)$ may be attainable under appropriate conditions.

**Conjecture 18.** *For any sequence of $\mathcal{H}_\ell$-adapted policies $\{\mu_\ell\}_{\ell=1}^L$,*

$$I(M; \mu_1, Y_{1,\mu_1}, \ldots, \mu_L, Y_{L,\mu_L}) = \tilde{O}(|\mathcal{S}||\mathcal{A}|).$$

The conjecture follows from the following conjecture about the mutual information between a Dirichlet random variable and categorical observations.

**Conjecture 19.** *Suppose that $p \sim \mathrm{Dirichlet}(\alpha)$ with $\alpha \in \Re^N$ and $\overline{\alpha} = \mathbf{1}^\top \alpha \geq 2$. Conditioned on $p$, $s_1, \ldots, s_n$ are drawn i.i.d. from $p$. Then,*

$$I(p; s_1, \ldots, s_n) \leq c \log(N) \log(n)$$

*for some fixed constant $c$.*

In Figure 1, we show simulation results that support the conjecture. We fix $\alpha = \left( \frac{2}{N}, \frac{2}{N}, \ldots, \frac{2}{N} \right) \in \Re^N$, and plot $I(p; s_1, \ldots, s_n)$ where $p \sim \mathrm{Dirichlet}(\alpha)$. We see in Figure 1(a) and 1(b) that the mutual information appears to grow logarithmically with $N$ and the number of observations.

We show how Conjecture 18 follows from Conjecture 19. Let $\overline{Y}_\ell = \{(r_{\ell sa1}, \ldots, r_{\ell sa\tau})_{sa}, (s'_{\ell sa1}, \ldots, s'_{\ell sa\tau})_{sa}\}$ be a collection of random variables where for each episode and each state-action pair, we sample $\tau$ rewards and $\tau$ next states from the true MDP. The trajectory $Y_{\ell,\mu_\ell}$ that the agent observes during episode $\ell$ is then a subset of $\overline{Y}_\ell$. We have

$$
\begin{aligned}
I(M; \mu_1, Y_{1,\mu_1}, \ldots \mu_L, Y_{L,\mu_L}) &\leq I(M; \mu_1, \ldots, \mu_L, \overline{Y}_1, \ldots, \overline{Y}_L) \\
&= I(M; \overline{Y}_1, \ldots, \overline{Y}_L) + I(M; \mu_1, \ldots, \mu_L \mid \overline{Y}_1, \ldots, \overline{Y}_L).
\end{aligned}
$$

Let $H(\cdot)$ denotes the Shannon entropy of a discrete random variable. Since $\mu_\ell$ is $\mathcal{H}_\ell$-adapted,

$$I(M; \mu_1, \ldots, \mu_L \mid \overline{Y}_1, \ldots, \overline{Y}_L) = \sum_{\ell=1}^{L} I(M; \mu_\ell \mid \overline{Y}_1, \ldots, \overline{Y}_L, \mu_1, \ldots, \mu_{\ell-1})$$

$$= \sum_{\ell=1}^{L} H(\mu_\ell \mid \overline{Y}_1, \ldots, \overline{Y}_L, \mu_1, \ldots, \mu_{\ell-1}) - H(\mu_\ell \mid \overline{Y}_1, \ldots, \overline{Y}_L, \mu_1, \ldots, \mu_{\ell-1}, M) = 0.$$

Therefore,

$$\begin{aligned}
I(M; \mu_1, Y_{1,\mu_1}, \ldots \mu_L, Y_{L,\mu_L}) &\leq I(M; \overline{Y}_1, \ldots, \overline{Y}_L) \\
&\leq \sum_{s,a} c \left( \log 2 + \log |\mathcal{S}| \right) \log T \\
&= O(|\mathcal{S}||\mathcal{A}| \log |\mathcal{S}| \log T).
\end{aligned}$$

We here establish a looser bound on the maximal information gain by any agent with an extra factor of $|\mathcal{S}|$.

**Lemma 20.** *Under assumption 2, for any sequence of $\mathcal{H}_\ell$-adapted policies $\{\mu_\ell\}_{\ell=1}^{L}$,*

$$I(M; \mu_1, Y_{1,\mu_1}, \ldots, \mu_L, Y_{L,\mu_L}) \leq (|\mathcal{S}| + 2)|\mathcal{S}||\mathcal{A}| \log \left( 1 + \frac{T}{|\mathcal{S}||\mathcal{A}|} \right).$$

The lemma relies on a simple bound on the information gain of a Dirichlet random variable under $n$ observations.

**Lemma 21.** *Suppose that $p \sim \mathrm{Dirichlet}(\alpha)$ with $\alpha \in \Re^N$ and $\overline{\alpha} = \mathbf{1}^\top \alpha \geq 2$. Conditioned on $p$, let $Y$ be drawn from $\mathrm{Multinomial}(n, p)$. Let $y$ be a realization of $Y$. Then,*
$$h(p) - h(p|Y = y) \leq N \log(\overline{\alpha} + n).$$

*Proof.* By Lemma 15 and Lemma 16,

$$h(p) - h(p|Y = y)$$

$$= \left( \sum_{j=1}^{N} \sum_{k=0}^{y_j - 1} (\psi(\alpha_j + k + 1) - \log(\alpha_j + k)) \right) - \left( \sum_{k=0}^{n-1} (\psi(\overline{\alpha} + k + 1) - \log(\overline{\alpha} + k)) \right)$$

$$- \sum_{j=1}^{N} \sum_{k=0}^{y_j - 1} \frac{1}{\alpha_j + k} + N \sum_{k=0}^{n-1} \frac{1}{\overline{\alpha} + k}$$

$$\leq \frac{1}{2} \sum_{j=1}^{N} \sum_{k=0}^{y_j - 1} \frac{1}{\alpha_j + k} - \sum_{j=1}^{N} \sum_{k=0}^{y_j - 1} \frac{1}{\alpha_j + k} + N \sum_{k=0}^{n-1} \frac{1}{\overline{\alpha} + k} \leq N \sum_{k=0}^{n-1} \frac{1}{\overline{\alpha} + k} \leq N \log(1 + n).$$

$\square$

*Proof of Lemma 20.* Let $n_{sa}$ denote the total number of times we have visited $(s, a)$ by the end of $L$ episodes. We have

$$\begin{aligned}
I(M; \mu_1, Y_{1,\mu_1}, \ldots, \mu_L, Y_{L,\mu_L}) &= h(M) - h(M|\mu_1, Y_{1,\mu_1}, \ldots, \mu_L, Y_{L,\mu_L}) \\
&\leq \mathbb{E} \left[ \sum_{s,a} 2 \log(1 + n_{sa}) + |\mathcal{S}| \log(1 + n_{sa}) \right] \\
&= (|\mathcal{S}| + 2) \mathbb{E} \left[ \sum_{s,a} \log(1 + n_{sa}) \right] \\
&\leq (|\mathcal{S}| + 2)|\mathcal{S}||\mathcal{A}| \mathbb{E} \left[ \log \left( \frac{1}{|\mathcal{S}||\mathcal{A}|} \sum_{s,a} (1 + n_{sa}) \right) \right] \\
&= (|\mathcal{S}| + 2)|\mathcal{S}||\mathcal{A}| \log \left( 1 + \frac{T}{|\mathcal{S}||\mathcal{A}|} \right),
\end{aligned}$$

where the second inequality follows from Jensen's inequality. $\square$

## D    Factored MDPs

**Lemma 12.** *Let* $r_{\ell,t,i,x_i^R} \sim \mathrm{Bernoulli}(R(x_i^R)) \mid \mathcal{H}_\ell, \mathcal{H}_{\ell t}$ *and* $s'_{\ell,t,j,x_j^P} \sim P_j(x_j^P, \cdot) \mid \mathcal{H}_\ell, \mathcal{H}_{\ell t}$. *Then,*

$$\mathbb{P}_\ell \left( \left| R_i(x_i^R) - \mathbb{E}_\ell R_i(x_i^R) \right| \leq \Gamma^R \sqrt{\min_{\tilde{t},h_{\tilde{t}}} I_\ell(R_i; x_i^R, r_{\ell,\tilde{t},i,x_i^R} \mid \mathcal{H}_{\ell\tilde{t}} = h_{\tilde{t}})} \right) \geq 1 - \delta \qquad (4)$$

*and*

$$\mathbb{P}_\ell \left( \|P_j(x_j^P) - \mathbb{E}_\ell P_j(x_j^P)\|_1 \leq \Gamma_j^P \sqrt{\min_{\tilde{t},h_{\tilde{t}}} I_\ell(P_j; x_j^P, s'_{\ell,\tilde{t},j,x_j^P} \mid \mathcal{H}_{\ell\tilde{t}} = h_{\tilde{t}})} \right) \geq 1 - \delta \qquad (5)$$

*for all* $s, a$ *such that* $\mathbf{1}^\top \alpha_{\ell,i,x_i^R}^R \geq \tau - 1$ *and* $\mathbf{1}^\top \alpha_{\ell,j,x_j^P}^P \geq \tau - 1$, *respectively, with*

$$\Gamma^R = 2\sqrt{6 \log \tfrac{2}{\delta}} \quad \text{and} \quad \Gamma_j^P = 4\sqrt{6|\mathcal{S}_j| \log \tfrac{2}{\delta}}.$$

*Proof.* (4) follows from the same proof as for Lemma 10. To prove (5), note that by Lemma 17, if $p \sim \mathrm{Dirichlet}(\alpha)$ where $\alpha \in \Re^N$ and $\overline{\alpha} = \mathbf{1}^\top \alpha \geq 2$, and and if $v \in \{-1, 1\}^N$ is a fixed vector, we have

$$\mathbb{P} \left( (p - \mathbb{E}p)^\top v \geq 2\sqrt{\frac{2}{\overline{\alpha}} \log \frac{1}{\delta}} \right) \leq \delta.$$

Since $\|p - \mathbb{E}p\|_1 = \max_{v \in \{-1,1\}^N} (p - \mathbb{E}p)^\top v$, the union bound implies that with probability at least $1 - \delta$,

$$\|p - \mathbb{E}p\|_1 \leq 2\sqrt{\frac{2}{\overline{\alpha}} \log \frac{2^N}{\delta}} \leq 2\sqrt{\frac{2N}{\overline{\alpha}} \log \frac{2}{\delta}}.$$

Hence, similar to Lemma 10, if $\overline{\alpha}_{\ell,j,x_j^P}^P = \mathbf{1}^\top \alpha_{\ell,j,x_j^P}^P \geq \tau - 1$, we have

$$\begin{aligned}
1 - \delta &\leq \mathbb{P}_\ell \left( \|P_j(x_j^P) - \mathbb{E}_\ell P_j(x_j^P)\|_1 \leq 2\sqrt{\frac{2|\mathcal{S}_j|}{\overline{\alpha}_{\ell,j,x_j^P}^P} \log \frac{2}{\delta}} \right) \\
&\leq \mathbb{P}_\ell \left( \|P_j(x_j^P) - \mathbb{E}_\ell P_j(x_j^P)\|_1 \leq 4\sqrt{6|\mathcal{S}_j| \min_{\tilde{t},h_{\tilde{t}}} I_\ell(P_j; x_j^P, s'_{\ell,\tilde{t},j,x_j^P} \mid \mathcal{H}_{\ell\tilde{t}} = h_{\tilde{t}}) \log \frac{2}{\delta}} \right).
\end{aligned}$$

$\square$

We will now construct information-theoretic confidence sets around MDPs. Let $\overline{R}_{\ell,i}$ and $\overline{P}_{\ell,j}$ denote the posterior mean of the reward and transition functions. Define shorthands

$$I_\ell^{\min} \left( R_i(x_i^R) \right) \equiv \min_{\tilde{t},h_{\tilde{t}}} I_\ell \left( R_i; x_i^R, r_{\ell,\tilde{t},i,x_i^R} \mid \mathcal{H}_{\ell\tilde{t}} = h_{\tilde{t}} \right),$$

$$I_\ell^{\min} \left( P_j(x_j^P) \right) \equiv \min_{\tilde{t},h_{\tilde{t}}} I_\ell \left( P_j; x_j^P, s'_{\ell,\tilde{t},j,x_j^P} \mid \mathcal{H}_{\ell\tilde{t}} = h_{\tilde{t}} \right).$$

Let $\mathcal{M}_\ell$ be the set of MDPs $\tilde{M}$ such that

$$\left| \tilde{R}_i(x_i^R) - \overline{R}_{\ell,i}(x_i^R) \right| \leq \Gamma^R \sqrt{I_\ell^{\min} \left( R_i(x_i^R) \right)} \quad \forall i, x_i^R \text{ s.t. } \overline{\alpha}_{\ell,i,x_i^R}^R \geq \tau - 1$$

and

$$\|\tilde{P}_j(x_j^P) - \overline{P}_{\ell,j}(x_j^P)\|_1 \leq \Gamma^P \sqrt{I_\ell^{\min} \left( P_j(x_j^P) \right)} \quad \forall j, x_j^P \text{ s.t. } \overline{\alpha}_{\ell,j,x_j^P}^P \geq \tau - 1,$$

where

$$\Gamma^R = 2\sqrt{6 \log \frac{4D\tau}{\delta}} \quad \text{and} \quad \Gamma^P = 4\sqrt{6K \log \frac{4D\tau}{\delta}}. \qquad (9)$$

Then, by Lemma 12 and union bound, we have $\mathbb{P}_\ell(M \in \mathcal{M}_\ell) \geq 1 - \frac{\delta}{2}$. The corresponding UCB algorithm will have upper confidence functions $U_\ell(\mu) = \max_{\hat{M}_\ell \in \mathcal{M}_\ell} \overline{V}_\mu^{\hat{M}_\ell}$.

**Proposition 13.** *The Bayesian regret of Thompson sampling and UCB over L episodes is*

$$\mathbb{E}[\text{Regret}(L, \pi)] = \tilde{O}\left(m\tau\sqrt{(m+n)K\tau L\, I\left(M; \mu_1, Y_{1,\mu_1}, \ldots, \mu_L, Y_{L,\mu_L}\right)}\right).$$

*Proof.* Let $\hat{M}_\ell = (\{\hat{R}_{\ell,i}\}_{i=1}^m$ and $\{\hat{P}_{\ell,j}\}_{j=1}^n)$ be the parameters sampled by Thompson sampling for episode $\ell$. By the probability matching property of Thompson sampling, we have $\mathbb{P}_\ell(\hat{M}_\ell \in \mathcal{M}_\ell) \geq 1 - \frac{\delta}{2}$. Then, by probability matching and Lemma 8, we have

$$
\begin{aligned}
\mathbb{E}_\ell[\Delta_\ell] &= \mathbb{E}_\ell\left[\overline{V}_{\mu^*}^M - \overline{V}_{\mu_\ell}^M\right] \\
&= \mathbb{E}_\ell\left[\overline{V}_{\mu_\ell}^{\hat{M}_\ell} - \overline{V}_{\mu_\ell}^M\right] \\
&\leq \sum_{t=0}^{\tau-1}\mathbb{E}_\ell\left[\mathbf{1}_{M,\hat{M}_\ell \in \mathcal{M}_\ell}\left(\left(\hat{R}_\ell(x_{\ell,t}) - R(x_{\ell,t})\right) + \left(\hat{P}_\ell(x_{\ell,t}) - P(x_{\ell,t})\right)^\top V_{\mu_\ell,t+1}^{\hat{M}_\ell}\right)\right] + \delta m\tau \\
&\leq \sum_{t=0}^{\tau-1}\mathbb{E}_\ell\left[\mathbf{1}_{M,\hat{M}_\ell \in \mathcal{M}_\ell}\left(\left|\hat{R}_\ell(x_{\ell,t}) - R(x_{\ell,t})\right| + m\tau\left\|\hat{P}_\ell(x_{\ell,t}) - P(x_{\ell,t})\right\|_1\right)\right] + \delta m\tau.
\end{aligned}
$$

We bound the deviation of reward and transition separately. We have

$$
\begin{aligned}
&\mathbb{E}_\ell\left[\mathbf{1}_{M,\hat{M}_\ell \in \mathcal{M}_\ell}\left|R(x_{\ell,t}) - \overline{R}_\ell(x_{\ell,t})\right|\right] \\
\leq\ &\mathbb{E}_\ell\left[\mathbf{1}_{M,\hat{M}_\ell \in \mathcal{M}_\ell}\sum_{i=1}^m\left|R_i(x_{\ell,t}[Z_i^R]) - \overline{R}_{\ell,i}(x_{\ell,t}[Z_i^R])\right|\right] \\
\leq\ &\mathbb{E}_\ell\left[\sum_{i=1}^m\sum_{x_i^R}\mathbf{1}(x_{\ell,t}[Z_i^R] = x_i^R)\Gamma^R\sqrt{I_\ell^{\min}\left(R_i(x_i^R)\right)}\right] + \mathbb{E}_\ell\left[\sum_{i=1}^m\mathbf{1}\left(\overline{\alpha}_{\ell,i,x_{\ell,t}[Z_i^R]}^R < \tau - 1\right)\right] \\
\leq\ &\mathbb{E}_\ell\left[\sum_{i=1}^m\Gamma^R\sqrt{I_{\ell t}(R_i; x_{\ell,t}, r_{\ell,t+1,i})}\right] + \mathbb{E}_\ell\left[\sum_{i=1}^m\mathbf{1}\left(\overline{\alpha}_{\ell,i,x_{\ell,t}[Z_i^R]}^R < \tau - 1\right)\right],
\end{aligned}
$$

where $\Gamma^R$ is defined in (9), and similarly for the deviation of $\hat{R}_\ell(x_{\ell,t})$.

To bound the deviation of transition functions, we will use Lemma 1 of [14], which allows us to bound

$$\|P(x) - \overline{P}(x)\|_1 \leq \sum_{j=1}^n\|P_j(x[Z_j^P]) - \overline{P}_j(x[Z_j^P])\|_1$$

for any two factored transition functions. We have

$$
\begin{aligned}
&\mathbb{E}_\ell\left[\mathbf{1}_{M,\hat{M}_\ell \in \mathcal{M}_\ell}\left\|P(x_{\ell,t}) - \overline{P}_\ell(x_{\ell,t})\right\|_1\right] \\
\leq\ &\mathbb{E}_\ell\left[\mathbf{1}_{M,\hat{M}_\ell \in \mathcal{M}_\ell}\sum_{j=1}^n\|P_j(x_{\ell,t}[Z_j^P]) - \overline{P}_j(x_{\ell,t}[Z_j^R])\|_1\right] \\
\leq\ &\mathbb{E}_\ell\left[\sum_{j=1}^n\sum_{x_j^P}\mathbf{1}(x_{\ell,t}[Z_j^P] = x_j^P)\Gamma^P\sqrt{I_\ell^{\min}\left(P_j(x_j^P)\right)}\right] + 2\mathbb{E}_\ell\left[\sum_{j=1}^n\mathbf{1}\left(\overline{\alpha}_{\ell,j,x_{\ell,t}[Z_j^P]}^P < \tau - 1\right)\right] \\
\leq\ &\mathbb{E}_\ell\left[\sum_{j=1}^n\Gamma^P\sqrt{I_{\ell t}(P_j; x_{\ell,t}, s_{\ell,t+1})}\right] + 2\mathbb{E}_\ell\left[\sum_{j=1}^n\mathbf{1}\left(\overline{\alpha}_{\ell,j,x_{\ell,t}[Z_j^P]}^P < \tau - 1\right)\right].
\end{aligned}
$$

where $\Gamma^P$ is defined in (9), and similarly for the deviation of $\hat{P}_\ell(x_{\ell,t})$.

Define

$$\epsilon_\ell = \delta m\tau + \mathbb{E}_\ell\left[\sum_{t=0}^{\tau-1}\sum_{i=1}^m\mathbf{1}\left(\overline{\alpha}_{\ell,i,x_{\ell,t}[Z_i^R]}^R < \tau - 1\right) + 2m\tau\sum_{j=1}^n\mathbf{1}\left(\overline{\alpha}_{\ell,j,x_{\ell,t}[Z_j^P]}^P < \tau - 1\right)\right].$$

Then, similar to the tabular case, we have

$$
\begin{aligned}
\mathbb{E}_\ell\left[\Delta_\ell\right] &\leq \sum_{t=0}^{\tau-1}\mathbb{E}_\ell\left[2\Gamma^R\sum_{i=1}^m\sqrt{I_{\ell t}(R_i;x_{\ell,t},r_{\ell,t+1,i})}+2m\tau\Gamma^P\sum_{j=1}^n\sqrt{I_{\ell t}(P_j;x_{\ell,t},s_{\ell,t+1})}\right]+\epsilon_\ell \\
&\leq 2(\Gamma^R+m\tau\Gamma^P)\sum_{t=0}^{\tau-1}\mathbb{E}_\ell\left[\sqrt{(m+n)\,I_{\ell t}\left(\{R_i\}_i,\{P_j\}_j;x_{\ell,t},\{r_{\ell,t+1,i}\}_i,s_{\ell,t+1}\right)}\right]+\epsilon_\ell \\
&\leq 2(\Gamma^R+m\tau\Gamma^P)\sqrt{(m+n)\tau\,I_\ell(M;\mu_\ell,Y_{\ell,\mu_\ell})}+\epsilon_\ell \\
&= \Gamma\sqrt{I_\ell(M;\mu_\ell,Y_{\ell,\mu_\ell})}+\epsilon_\ell,
\end{aligned}
$$

where

$$
\Gamma = 2(\Gamma^R+m\tau\Gamma^P)\sqrt{(m+n)\tau} = O\left(m\tau\sqrt{(m+n)K\tau\log\frac{D\tau}{\delta}}\right).
$$

Take $\delta=\frac{1}{L}$. The sum of errors

$$
\mathbb{E}\sum_{l=1}^L\epsilon_\ell \leq m\tau+\tau D_R+2m\tau^2 D_P.
$$

The regret bound for Thompson sampling then follows from Proposition 2.

For a UCB algorithm with upper confidence functions $U_\ell(\mu) = \max_{\hat{M}\in\mathcal{M}_\ell}\overline{V}_\mu^{\hat{M}}$, let $\mu_\ell = \arg\max_\mu U_\ell(\mu)$ and let $\hat{M}_\ell = \arg\max_{\hat{M}\in\mathcal{M}_\ell}\overline{V}_{\mu_\ell}^{\hat{M}}$. We have

$$
\mathbb{E}_\ell\left[\Delta_\ell\right]\leq\mathbb{E}_\ell\left[\mathbf{1}_{M\in\mathcal{M}_\ell}\left(\overline{V}_{\mu^*}^M-\overline{V}_{\mu_\ell}^M\right)\right]+\frac{1}{2}\delta m\tau\leq\mathbb{E}_\ell\left[\mathbf{1}_{M\in\mathcal{M}_\ell}\left(\overline{V}_{\mu_\ell}^{\hat{M}_\ell}-\overline{V}_{\mu_\ell}^M\right)\right]+\frac{1}{2}\delta m\tau.
$$

The rest of the analysis follows similarly. $\qquad\square$

**Lemma 22.** *For any policy sequence, the information gain of a factored MDP over $L$ episodes is*

$$
I\left(\{R_i\}_{i=1}^m,\{P_j\}_{j=1}^n;\mu_1,Y_{1,\mu_1},\ldots,\mu_L,Y_{L,\mu_L}\right)\leq KD\log(1+T).
$$

*Proof.* The proof is similar to the proof of Lemma 20. Let $n_{i,x_i^R}^R$ and $n_{j,x_j^P}^P$ denote the final counts of the scope variables. We have

$$
\begin{aligned}
&I\left(\{R_i\}_{i=1}^m,\{P_j\}_{j=1}^n;\mu_1,Y_{1,\mu_1},\ldots,\mu_L,Y_{L,\mu_L}\right) \\
&\leq \mathbb{E}\left[\sum_{i=1}^m\sum_{x_i^R\in\mathcal{X}_i[Z_i^R]}\log\left(1+n_{i,x_i^R}^R\right)+\sum_{j=1}^n\sum_{x_j^P\in\mathcal{X}_j[Z_j^P]}|\mathcal{S}_j|\log\left(1+n_{j,x_j^P}^P\right)\right] \\
&\leq D_R\log\left(1+\frac{mT}{D_R}\right)+KD_P\log\left(1+\frac{nT}{D_P}\right) \\
&\leq KD\log(1+T).
\end{aligned}
$$

$\qquad\square$

Similar to the tabular case, we conjecture that a bound of order $\tilde{O}(D)$ may be attainable under appropriate conditions.