[Reviews · NeurIPS 2019]

Reviewer 1



- Originality: Although related ideas have been proposed in [1], here they are extended to more general results involving confidence bounds over rewards/observations, and are applied to more general multi-step RL settings. - Quality: The analysis is quite thorough and seems correct, although I have not checked the proofs in detail. - Clarity: The paper is well-written for a theory audience, but fairly dense and requires careful reading, which I think will limit its reach to the wider community. It would be great to make it more accessible to a more general audience, as the ideas it contains are fairly intuitive at their core. One suggestion would be to include illustrative figures to convey the general intuition, for example for the case of the linear-Gaussian bandit, since the confidence sets have a natural geometric interpretation in terms of the variance of the posterior. - Significance: I think these results have the potential to be significant, because they give general tools for computing regret bounds for the RL algorithms applied in different scenarios. The analyses of the examples given (linear bandits, MDPs, factored MDPs) essentially all follow a recipe made possible by the results relating the confidence bounds to the regret. Specifically, they are: 1) Construct a confidence interval based on the mutual information using the characteristics of the problem at hand (linearity/Gaussian noise assumptions for the bandit, specific forms of the prior for MDPs) 2) Bound the sum of the information gain. Combining these two then gives a regret bound. It is possible that this same approach could be applied to a broad class of problems other than the ones considered here. ### Minor Comments: - Line 26: please add references for "Most analyses..." - Line 69: should "a" be "A_\ell" here? This notation is used beyond the first step later in the paper, but here is says "Y_{1, a}". - Section 4 and elsewhere: It seems the \theta variable is overloaded at different places. For example, Line 111 \theta refers to the true model but elsewhere it refers to the random variable. Please use \theta^\star or something similar when referring to the true model.

Reviewer 2



## Summary: This paper builds upon the information-theoretic analysis of Thompson sampling in Russo and Van Roy (JMLR'16). The authors develop a per-stage bound on the expected regret in terms of the mutual information between a parameter describing an unknown environment model and the current action and observation outcome. The per-stage regret bound can, in turn, be used to obtain a bound on the total regret. The authors, also, show that the per-stage regret bound is related to a bound on the deviation of the reward function from its mean. The proposed bounds are computed explicitly in the cases of linear bandits, tabular MDPs, and factored MDPs. ## Comments: The paper offers novel regret bounds for Thompson sampling and UCB algorithms but the key idea is mostly a straightforward extension of Russo and Van Roy (JMLR'16). The detailed derivations for all three cases (linear bandits, tabular MDPs and factored MDPs) and the explicit computation of the Gamma term for common probability distributions are particular strengths of the paper. A weakness of the paper is that the theoretical results use standard properties of mutual information and Gaussian/Dirichlet/Bernoulli distributions and hence lead to very general and very loose bounds. It is interesting to see the order of the regret bounds for the MDP setting but the bounds seem quite loose and not very useful for developing new algorithms. There are some problems with the notation in the paper. The use of distribution, density, and measure is incorrect. In the first definition of KL divergence, P is used as a probability measure. In the second, definition of mutual information P is used with a strange notation which comes from ref. [10] but is not introduced in this paper. Without a proper definition, the expression P(X,Y) is meaningless since the input for a measure function is a set. New notation \mathbb{P} is introduced in Sec. 3.2; \ldots is missing in the history definition. There is a switch from \ell to t to denote the time steps in the middle of the paper, yet H_\ell is still used as to denote the observation history. ## Update after rebuttal The authors acknowledge that the notation should be improved and offer to add simulated examples of the information gain and regret bounds. This would certainly strengthen the paper and address my main concerns.

Reviewer 3



This paper integrates information-theoretic concepts into the design and analysis of optimistic algorithms and Thompson sampling. By making a connection between information-theoretic quantities and confidence bounds to obtain results that relate the per-period performance of the agent with its information gain about the environment, thus explicitly characterizing the exploration-exploitation tradeoff. The resulting cumulative regret bound depends on the agent’s uncertainty over the environment and quantifies the value of prior information. This paper show applicability of this approach to several environments, including linear bandits, tabular MDPs, and factored MDPs. These examples demonstrate the potential of a general information-theoretic approach for the design and analysis of reinforcement learning algorithms.

[Author Response · NeurIPS 2019]

We would like to thank all the reviewers for their time and helpful feedback.

All reviewers mention that the current version of the paper is a little dense. We will work on improving clarity and
accessibility and try to convey more intuition in the main text. We would like to thank reviewers 1 and 2 for their helpful
suggestions on how to make the paper more accessible.

We also plan to add more simulation results in the final version. To reviewer 1, we will add simulation results for
the conjecture. To reviewer 2, we completely agree that it would be very interesting to see simulated examples of
information gain and regret bounds. For the linear-Gaussian bandit, the regret bound derived in the paper is comparable
to the best bounds in the literature up to logarithmic factors. It will be interesting to see the simulated total information
gain and plot the corresponding regret bounds. For MDPs, we suspect that the upper bounds on total information gain
are loose and the actual information gain might be much smaller.

Several reviewers ask about how the paper might lead to new algorithms. This is definitely an important direction for
future research. As a starting point, it may be interesting to experiment with UCB algorithms that use information
gain in their upper confidence functions. They come with regret guarantees for problems discussed in the paper, and it
would be interesting to see how they perform in practice. UCB algorithms that are studied previously usually construct
confidence sets around the model parameter based on past observations, and they do not consider how much information
would be revealed about the model when we see a new observation. By explicitly taking into account information
gain, these new UCB algorithms tend to give a higher bonus to actions that reveal more information about the system
compared to existing UCB algorithms. It would be interesting to see whether these new algorithms would work better
for some classes of problems.

Finally, we will work on clarifying the notations and proofs.

Below are additional responses to individual reviewers.

**Reviewer 1**

– *Line 26: please add references for "Most analyses.."*
Thanks for pointing it out. We will add the references.

– *Line 69: should $a$ be $A_\ell$ here?*
We use $A_\ell$ to denote a possibly random action selected by the algorithm, while $a$ is used to denote some fixed action.
We will clarify this in the final version.

– *It seems the $\theta$ variable is overloaded at different places. For example, Line 111 $\theta$ refers to the true model but*
*elsewhere it refers to the random variable.*
Under our Bayesian framework, the true model is a random variable. We will emphasize this in the final version.

**Reviewer 2**

– *The paper offers novel regret bounds for Thompson sampling and UCB algorithms but the key idea is mostly a*
*straightforward extension of Russo and Van Roy (JMLR'16)...*
It is true that our main idea bears similarity to Russo and Van Roy (JMLR'16), but a major difficulty of applying
their approach to complex environments is that it is unclear how to analyze the mutual information between optimal
actions and observations, as optimal actions themselves are fairly complicated objects in complex environments. Our
approach considers a more natural information gain between the model and observations, which sidesteps their issue
and allows us to obtain information-theoretic bounds for more complex problems like MDPs and factored MDPs that
are not achieved in their paper.

– *There are some problems with the notation in the paper...*
We will clarify our notation on probability measures. We use $T$ to denote the total number of time periods for the
linear bandit problem as it is more consistent with other bandit literatures, but I see how it can cause confusions and
we will make it clear in the final version.

– *...presenting less material (e.g., leave the factored MDPs for the appendix)...*
We include factored MDPs in the main text as an example to demonstrate that our approach is able to handle complex
structured environments that could not be handled using tools from previous work such as Russo and Van Roy
(JMLR'16).

**Reviewer 3**

– *...more exhaustive descriptions of the proof will help the readers and the reviewer to understand the paper...*
We will polish the proofs for the final version.

[Meta-Review · NeurIPS 2019]

The paper extends Russo and Van Roy (JMRL2016) work to provide information-theoretical analysis of Thompson sampling and UCB-like algorithms in more general setting. The three reviewers acknowledge the contributions, and the potential impact of connecting information-theoretical concepts to the design of algorithms. Reviewers have suggested ways to improve the manuscript. The authors should follow these directions, and in particular fix notations, include simulation results, and provide explanations about proofs when necessary. The contributions in this paper are “methodological”, i.e., it proposes a framework to analyze the regret of certain algorithms. When applying the method to examples, the authors do not discuss existing results (what is known about the regret in linear bandits, tabular MDPs, …?). We encourage the authors to include a more detailed related work. On the examples presented in the paper, the proposed method does lead to regret upper bounds matching those of state-of-the-art algorithms (e.g. the MDP tabular case – unless the conjecture mentioned by the authors is true). It seems important to mention this gap.